# Identification of plants' functional counterpart of the metazoan mediator of DNA Damage checkpoint 1

Zdravko J Lorković [1✉], Michael Klingenbrunner [1,2], Chung Hyun Cho [1] & Frédéric Berger [1✉]

## Abstract

**Induction of DNA damage triggers rapid phosphorylation of the histone H2A.X (γH2A.X). In animals, mediator of DNA damage checkpoint 1 (MDC1) binds γH2A.X through a tandem BRCA1 carboxyl-terminal (tBRCT) domain and mediates recruitment of downstream effectors of DNA damage response (DDR). However, readers of this modification in plants have remained elusive. We show that from the *Arabidopsis* BRCT domain proteome, BCP1-4 proteins with tBRCT domains are involved in DDR. Through its tBRCT domain BCP4 binds γH2A.X in vitro and localizes to DNA damage-induced foci in an H2A.X-dependent manner. BCP4 also contains a domain that interacts directly with NBS1 and thus acts as a functional counterpart of MDC1. We also show that BCP1, that contains two tBRCT domains, co-localizes with γH2A.X but it does not bind γH2A.X suggesting functional similarity with human PAXIP1. A phylogenetic analysis supports that PAXIP1 and MDC1 in metazoa and their plant counterparts evolved independently from common ancestors with tBRCT domains. Collectively, our study reveals missing components and provides mechanistic and evolutionary insights into plant DDR.**

**Keywords** BCP4; BRCT Domain; DNA Damage Response; Histone H2A.X; MDC1
**Subject Categories** DNA Replication, Recombination & Repair; Plant Biology

## Introduction

The genome is constantly exposed to endogenous or exogenous sources of genotoxic agents, which cause DNA damage. Among different types of DNA damage, DNA double-strand breaks (DSBs) are the most deleterious as they can cause genome rearrangements if not accurately repaired. To prevent such harmful consequences, the DNA damage response (DDR) pathway senses DNA damage and initiates a signaling cascade by attracting sensor, transducer, mediator, and effector repair proteins to the site of DNA damage (Ciccia and Elledge, 2010).

Major protein factors contributing to signaling and repair of damaged DNA have been characterized. The earliest step in DDR is recruitment of the MRE11-RAD50-NBS1 (MRN) complex which senses and locates sites of DSBs. Stalled replication forks are sensed by replication protein A, which binds to ssDNA generated at the site of stalled replication forks (Ciccia and Elledge, 2010; Coster and Goldberg, 2010). These initial events lead to the recruitment of ATM and ATR kinases that phosphorylate a plethora of proteins acting downstream of DNA damage sensing (Lanz et al, 2019). Among these, the histone variant H2A.X is rapidly phosphorylated at its C-terminal SQ motif (SQEF/Y) and forms numerous γH2A.X foci throughout the nucleus (Rogakou et al, 1998; Paull et al, 2000; Turinetto and Giachino, 2015). These early events during DDR are conserved in fungi, animals, and plants (Nisa et al, 2019; Yoshiyama et al, 2013).

In mammals, γH2A.X promotes DDR by binding mediator of DNA damage checkpoint protein 1 (MDC1) (Stewart et al, 2003), which acts as a scaffold for DDR effectors (Coster and Goldberg, 2010). DDR effectors often contain BRCA1 C-terminus (BRCT) domains, which are present either as a single domain or two closely spaced tandem domains (tBRCT). tBRCT domains represent a distinct class of BRCT domains that bind phosphorylated target proteins (Manke et al, 2003; Yu et al, 2003). MDC1 interacts with γH2A.X through its C-terminal tBRCT domain (Stucki et al, 2005). In addition to the tBRCT domain, human MDC1 harbors four additional interaction domains/motifs. The N-terminal forkhead-associated (FHA) domain interacts with additional MDC1 molecules and RAD51 (Zhang et al, 2005). The FHA domain is followed by SDTD motifs which, when phosphorylated by casein kinase II, bind NBS1 of the MRN complex (Chapman and Jackson, 2008; Melander et al, 2008; Spycher et al, 2008; Wu et al, 2008). Binding of MRN complexes to MDC1 recruits additional ATM molecules, which further phosphorylate neighboring H2A.X proteins. Thus, MDC1 facilitates the spread of the DDR signal as far as one megabase up- and downstream of the initial DSB (Rogakou et al, 1999; Lou et al, 2006; Turinetto and Giachino, 2015). ATM also phosphorylates the TQxF motifs of MDC1, allowing recruitment of ring finger protein 8 (RNF8) (Kolas et al, 2007; Mailand et al, 2007), which ubiquitylates H2A/H2A.X proteins in its vicinity. These ubiquitylation marks are recognized by ring finger protein 168 which results in further ubiquitylation of H2A/H2A.X and chromatin relaxation around DNA damage site and provides access for repair factors to DNA (Gatti et al, 2015; Ferrand et al, 2020). The PST repeats of MDC1 directly interact with chromatin, and this is critical for TP53BP1 recruitment to DNA damage sites and cell survival in the absence of H2AX

[1]Gregor Mendel Institute (GMI), Austrian Academy of Sciences, Vienna Biocenter (VBC), Vienna, Austria. [2]Present address: Department of Molecular Life Sciences, University of Zurich, Zurich, Switzerland. ✉E-mail: zdravko.lorkovic@gmi.oeaw.ac.at; frederic.berger@gmi.oeaw.ac.at

(Salguero et al, 2019). TP53BP1 is a tBRCT domain protein that in concerted action with RNF8 is necessary for the recruitment of another tBRCT domain-containing DDR effector, PAXIP1 (Jowsey et al, 2004; Muñoz et al, 2007 Gong et al, 2009). This complex network of interactions between DDR effectors and associated chromatin modifications are necessary for the recruitment of repair machineries and influence the choice of DNA repair pathway (Ciccia and Elledge, 2010; Escribano-Diaz and Durocher, 2013; Ferrand et al, 2020; Mirman and de Lange, 2020; Xu and Xu, 2020).

Hence, the BRCT domain is a hallmark of proteins participating in DDR (Callebaut and Mornon, 1997; Leung and Glover, 2011). Among metazoan DDR mediator/effector proteins that contain BRCT domains (BRCA1, BARD1, TP53BP1, MDC1, TOPBP1, PAXIP1, and NBS1) only BRCA1 (Lafarge and Montané 2003), BARD1 (Reidt et al, 2006), and NBS1 (Akutsu et al, 2007; Waterworth et al, 2007) were unambiguously identified in plant lineage (Yoshiyama et al, 2013). Thus, how phosphorylated H2A.X is recognized and how this signal mediates the recruitment of DNA damage repair machineries in plant cells remains elusive. To identify the readers of phosphorylated H2A.X, we used a minimal complement of human BRCT proteins (Woods et al, 2012) to identify a complete set of *Arabidopsis* BRCT domain-containing proteins. Among the 21 BRCT domain proteins identified, we identified two potential functional counterparts of MDC1 and PAXIP1 and characterized this missing link between DDR signaling and DNA damage repair in plants. Based on a phylogenetic analysis and functional validations, we propose a potential evolutionary scenario for this complex family of proteins.

# Results

## BRCT domain proteome in plants

Using a combination of search strategies, we identified 21 *Arabidopsis* proteins with at least one BRCT domain (Fig. 1A). Of the 21 proteins identified, 11 were homologs of human proteins with reported BRCT domains (BRCA1, BARD1, XRCC1, REV1, PARP1, LIG4, RFC1, DPOLL, CTDP1, NBS1, and PES1) as revealed by clustering of *Arabidopsis* and human homologs on the unrooted phylogenetic tree (Fig. 1B), which also reflected homology beyond the BRCT domains. Overall, the domain organization of *Arabidopsis* BRCT proteins mirrored that of the human homologs, with some exceptions: AtPARP1 contains two Zinc fingers at the N-terminus that are missing in the human protein; the AtBRCA1 and AtBARD1 orthologs have extended PHD fingers in front of the BRCT domain (Fig. 1A); the AtCPL4, a CTDP1-like phosphatase contains a BRCT domain absent in the human ortholog; and the absence of a BRCT domain in the AtPARP4 although this domain is present in AtPARP3. In *Arabidopsis*, we could not find proteins with combinations and order of domains present in the human proteins MCPH1, POLM, DBF4B, LIG3, DNTT, PAXIP1, ANKRD32, TP53BP1, TOPBP1, and MDC1. The absence of TP53BP1 is not surprising as p53 itself is not present in plant lineage, rather its function is replaced by the plant-specific protein SOG1 (Yoshiyama et al, 2009; Yoshiyama et al, 2013). The function of TOPBP1, with eight BRCT domains, may be mediated by AtMEI1, which contains only five BRCT domains (Mathilde et al, 2003). Two additional *Arabidopsis*

proteins containing two BRCT domains at the N-terminal half of the protein, which we named TOPBP-LIKE1 (TOPBPL1) and TOPBP-LIKE2 (TOPBPL2), showed the best homology to TOPBP1 and clustered together with AtMEI1 and human TOPBP1 (Fig. 1B). The Sumo targeted ubiquitin ligase 2 (STUbL2), with one BRCT domain at the N-terminus, a RING finger, and an unannotated PHD finger at the C-terminus (Fig. 1A), has been identified as a suppressor of heterochromatin over-replication caused by the loss of the histone K27 methyltransferases ATXR5 and 6 (Hale et al, 2016). STUbL2 is most closely related to the human RNF8. Instead of a BRCT domain, RNF8 has an FHA domain at the N-terminus and a RING finger at the C-terminus. The FHA domain predominantly binds phospho-threonine (Almawi et al, 2017), while the BRCT domain preferentially binds phosphoserine (Manke et al, 2003; Yu et al, 2003; Leung and Glover, 2011), suggesting that the two proteins have distinct targets.

Four tBRCT domain proteins, BCP1-BCP4, were also identified in *Arabidopsis* while this work was in progress (Vladejić et al, 2022). BCP2, with a histone acetyltransferase domain, appears to be a plant-specific BRCT domain protein of unknown function. The three other BRCT domain proteins (BCP1, BCP3, and BCP4) did not reveal an obvious domain organization comparable to any human protein. BCP1, BCP3, and BCP4 were conserved in all plant lineages analyzed, with the separation of clades corresponding to algae, non-flowering land plants, basal angiosperms, monocots, and dicots (Fig. EV1A,B). BCP3 and BCP4 appear to be paralogs that arose from genome duplication events, as shown by their same exon–intron arrangements (Fig. EV1C). Algae, non-flowering land plants, gymnosperms, and basal angiosperms contain only one BCP3/4 protein, suggesting that this gene duplication took place with the appearance of eudicots.

BCP1 has two tBRCT domains positioned at the N- and C-terminus and a so far unrecognized C-terminal PHD finger which is present in all plant lineages except Brassicaceae (Figs. EV1A and EV2A). BCP3 and BCP4, which both have a tBRCT domain at the C-terminus, showed limited homology with tBRCT domains of human MDC1 and PAXIP1 proteins (Fig. 1B,C). On an unrooted phylogenetic tree, BCP3 and BCP4 clustered with MDC1 and PAXIP1 when we used either the entire protein sequences (Fig. 1B) or only the tBRCT domains (Fig. 1D). However, BCP3 and BCP4 contained only two BRCT domains in contrast to the six BRCT domains present in PAXIP1 (Jowsey et al, 2004) and did not contain long stretches of glutamine (Figs. 1A and EV2B), suggesting that they are not PAXIP1 homologs. Altogether our analysis suggested that BCP3 and BCP4 might be more functionally related to MDC1 and recognize γH2A.X.

## DNA damage sensitivity and DDR of *BCP* mutants

To analyze the potential role of BCPs in DDR, we obtained T-DNA insertion mutants of *BCP1-4* (Fig. EV1D) and measured their sensitivity to DNA damage using the true leaf assay (Rosa and Mittelsten Scheid, 2014). Wild-type plants and mutants deprived of H2A.X (*hta3 hta5*) or H2A.W.7 (*hta7*) with demonstrated DNA damage sensitivity (Lorković et al, 2017) were used as controls. In contrast to recently published data (Vladejić et al, 2022), all *bcp* alleles were sensitive to the zeocin that causes DSBs (Fig. 2A). The mutant *bcp3* was not as sensitive as *bcp1*, *bcp2*, or *bcp4* (Fig. 2A), presumably because the T-DNA insertion in *bcp3* is located in the 5'-UTR (Fig. EV1D) and likely did not completely knock out BPC3.

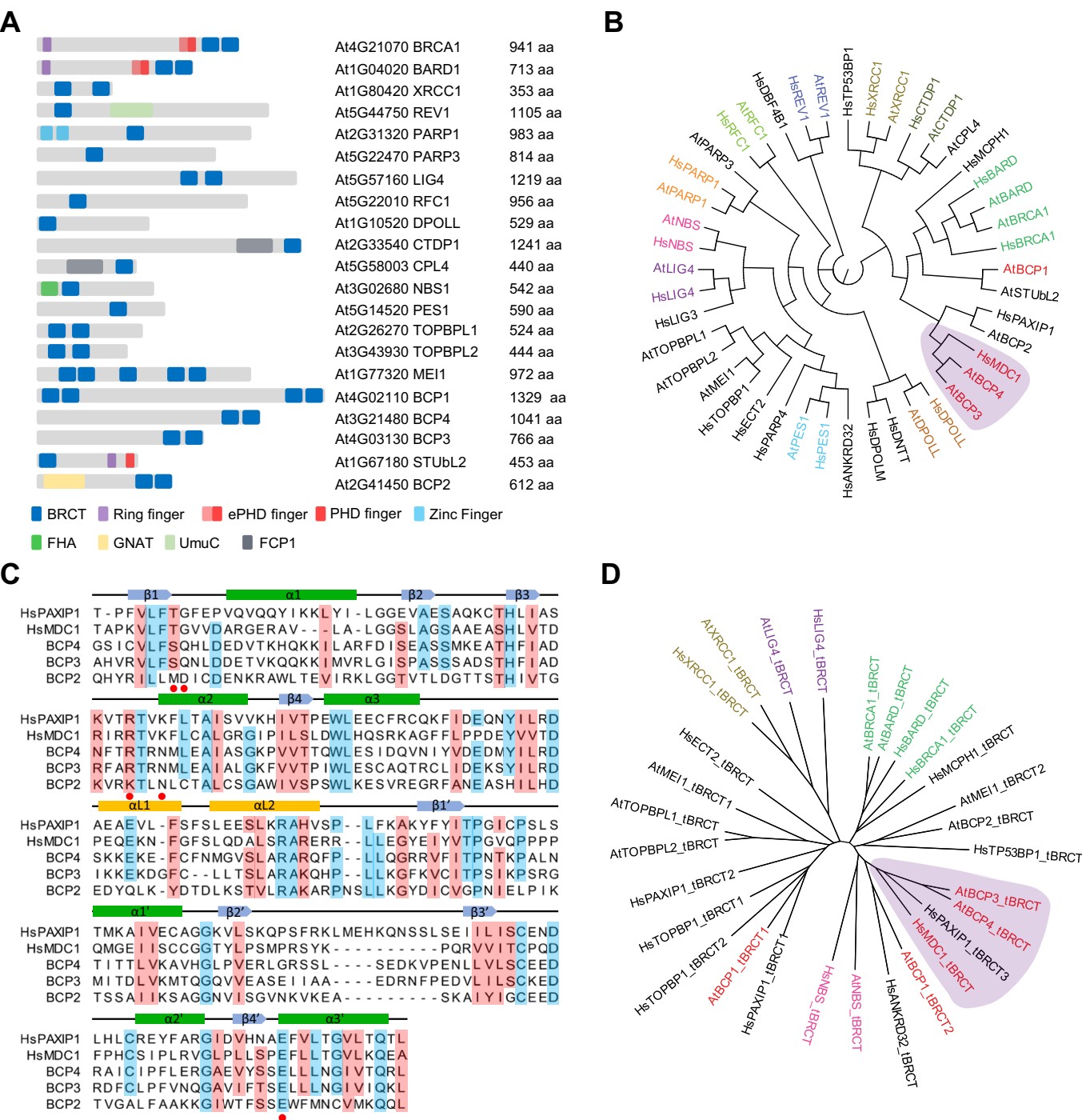

**Figure 1. Identification of *Arabidopsis* proteins with BRCT domains.**

(**A**) Schematic representation of 21 *Arabidopsis* proteins containing BRCT domains. The gene codes, protein names, and length in amino acids (aa) are indicated. (**B**) Human and *Arabidopsis* BRCT domain proteins were aligned with MUSCLE and a maximum likelihood circular cladogram was generated with CLC Genomics Workbench ver. 11.0. (**C**) Sequence alignment of *Arabidopsis* BCP3 and BCP4 and human MDC1 and PAXIP1 tBRCT domains. For alignment, tBRCT3 of PAXIP1 was used. Residues conserved in at least four proteins are shaded light blue and those similar in at least four proteins are shaded light red. Structural elements of domains are indicated on the top of the alignment according to Stucki et al (2005). Amino acids involved in MDC1 interaction with γH2A.X are indicated with red dots. (**D**) All human and *Arabidopsis* tBRCT domains were aligned as in (**B**) and a maximum likelihood tree illustrating amino acid sequence conservation between tBRCT domains was generated with CLC Genomics Workbench ver. 11.0. (**B, D**) Human and *Arabidopsis* proteins with high sequence homology share the same color code. *Arabidopsis* proteins and tBRCT domains clustering with human MDC1 and PAXIP1 are indicated in red and shaded in purple.

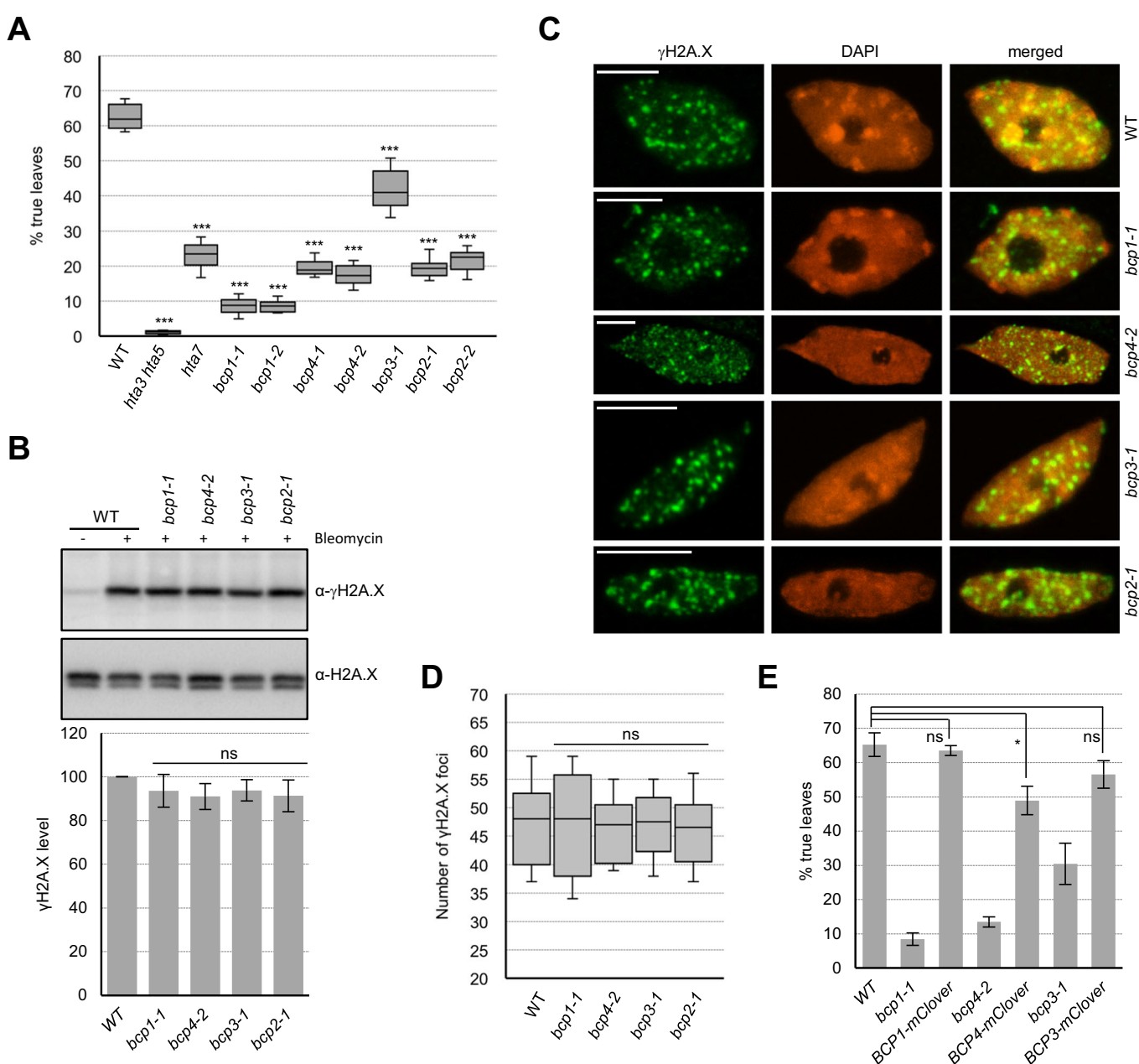

**Figure 2. Analysis of *bcp* mutants.**

(A) DNA damage sensitivity of *BCP* mutant lines assessed by true leaf development assay (*n* = 13 biological replicates, each with 64 seeds). DNA damage-sensitive mutants of H2A.X (*hta3 hta5*) and H2A.W.7 (*hta7*) histone variants were used as controls. Data information: Data are represented as a box plot with median and interquartile range (box) and minimal and maximal values (whiskers). ***P ≤ 0.0001 (two-tailed paired Student's *t* test). (B) Analysis of γH2A.X levels in *bcp* mutant seedlings after induction of DNA damage for 2 h. Representative western blots for γH2A.X and H2A.X in BCP mutants. Quantified γH2A.X levels normalized to the total H2A.X (*n* = 6, three biological replicates with two technical repeats). (C) Analysis of γH2A.X foci in nuclei of WT and *bcp* mutants. Data information: Maximum intensity projection images from Z-stacks of representative nuclei are shown. Scale bars represent 5 µm. (D) Quantification of γH2A.X foci number in *bcp* mutants. For each genetic background *n* = 8 nuclei were used. Data are represented as a box plot with median and interquartile range (box) and minimal and maximal values (whiskers). Data information: P > 0.05, ns (nonsignificant). (two-tailed paired Student's *t* test). (E) DNA damage sensitivity of BCP-mClover3 complementing lines (*n* = 6 biological replicates, each with 64 seeds). Data information: In (B, E), data are presented as mean ± SD. ***P ≤ 0.0001; **P ≤ 0.001; *P ≤ 0.01; P > 0.01, ns (nonsignificant). (two-tailed paired Student's *t* test). In (A, E) Seeds were germinated on medium containing 20 µg/ml of zeocin and true leaf development was scored 12 days after germination. Source data are available online for this figure.

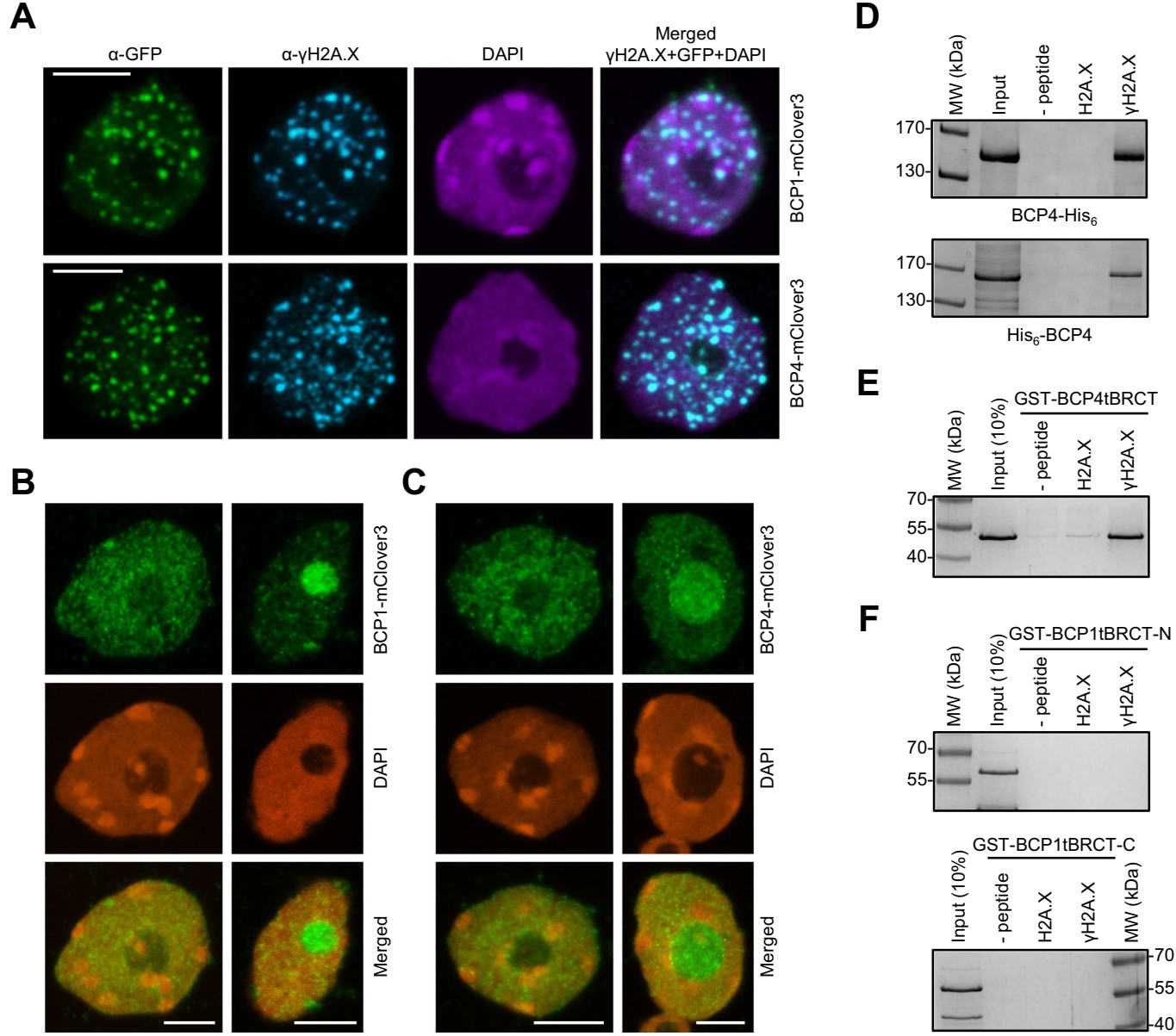

**Figure 3. BCP4 and MDC1 share similar properties.**

(A) BCP1 and BCP4 co-localize with γH2A.X foci. Immunostaining of nuclei from bleomycin-treated seedlings expressing BCP1-mClover3 and BCP4-mClover3 in *bcp1* and *bcp4* mutants, respectively. (B, C) BCP1 and BCP4 foci formation is dependent on H2A.X Immunostaining of nuclei from bleomycin-treated seedlings expressing BCP1-mClover (B) and BCP4-mClover (C) in *bcp1 hta3 hta5* and *bcp4 hta3 hta5* genetic backgrounds, respectively. (D) BCP4 binds phosphorylated H2A.X C-terminal peptide. Affinity pull-down with N- and C-terminally His-tagged BCP4 and biotinylated H2A.X peptides. (E) Affinity pull-down with GST-tagged tBRCT domain of BCP4. (F) The tBRCT domains of BCP1 do not bind phosphorylated H2A.X C-terminal peptide. Affinity pull-down with GST-tagged tBRCT domains and biotinylated H2A.X peptides. Data information: In (A–C), maximum intensity projection images from Z-stacks are shown and scale bars represent 5 μm. Proteins were analyzed on 10% (D) or 12% (E, F) SDS-PAGE and gels were stained with Coomassie blue. Source data are available online for this figure.

We also assessed DNA damage response in *bcp* mutants by observing the increased levels of γH2A.X in response to a 2-h treatment of seedlings with bleomycin. In all mutant alleles, comparable levels of γH2A.X were observed compared to WT seedlings (Fig. 2B). Nuclei from bleomycin-treated seedlings did not show an obvious difference in either the number or size of γH2A.X foci between the *bcp* mutants and WT (Fig. 2C,D), suggesting that BCP proteins are not required for either the initiation or propagation of γH2A.X.

We next complemented the loss of function mutants *bcp1*, *bcp3*, and *bcp4* with respectively BCP1-mClover3, BCP3-mClover3, and BCP4-mClover3 fusion proteins. Transgenic lines expressing mClover3-tagged genomic constructs were not sensitive to zeocin, confirming that the sensitivity in these mutant alleles was due to the absence of functional BCP proteins (Fig. 2E). Together, these data establish that BCP1-4 are required for proper DDR in *Arabidopsis*.

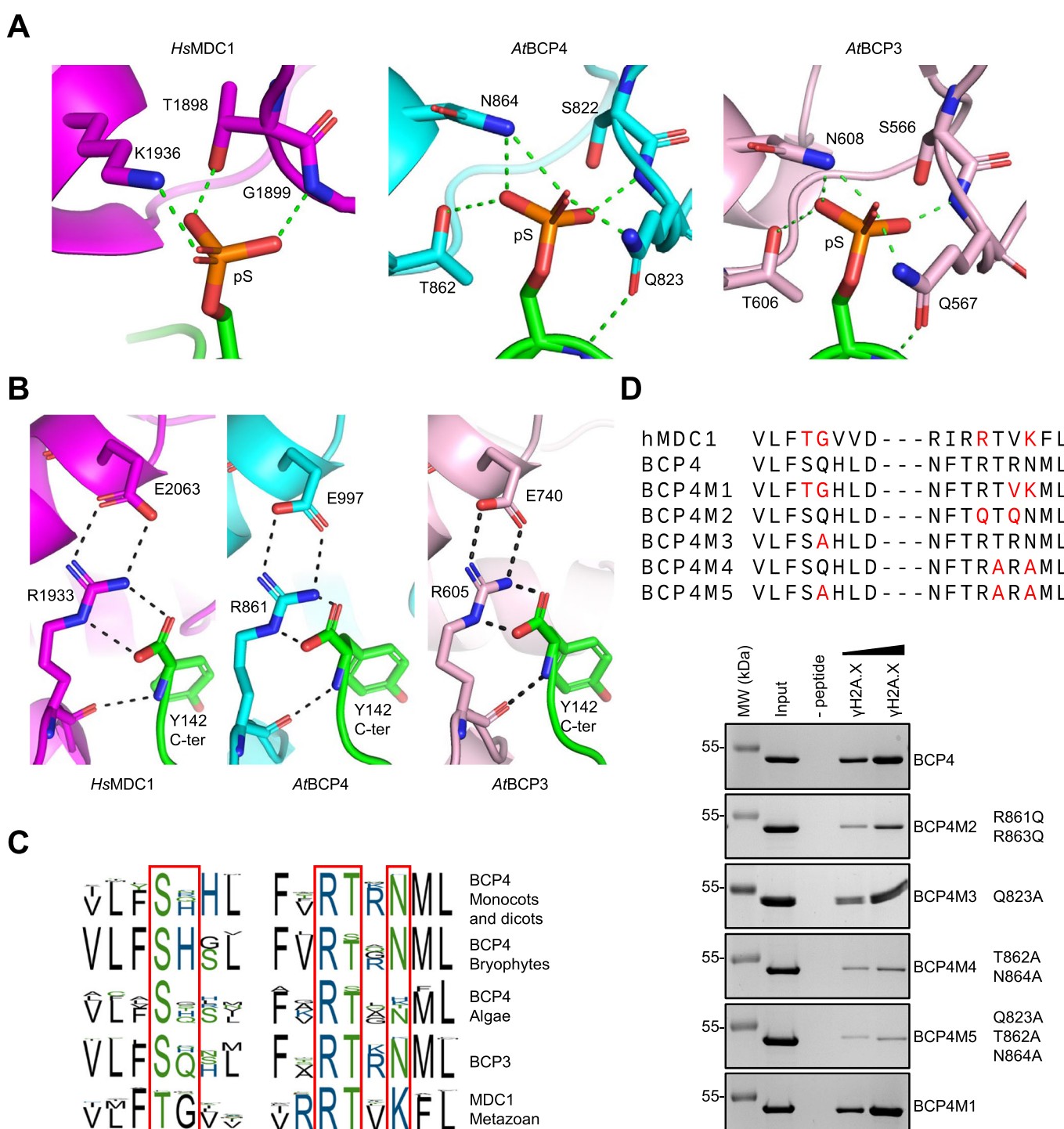

**Figure 4. BCP4 binds γH2A.X through a structurally conserved binding pocket.**

(A) Comparison of interactions of tBRCT domains of MDC1, BCP4, and BCP3 with phosphorylated C-terminal peptide of H2A.X. Published (MDC1; Stucki et al, 2005) and AlphaFold2 predicted (BCP3 and BCP4) contacts of indicated amino acids with pSer of H2A.X are indicated with dotted lines. (B) Contacts of the human Arg1933 with the C-terminal carboxyl group of H2A.X and Glu2063 are conserved in *Arabidopsis* BCP3 and BCP4. (C) Phosphoserine-binding consensus sequences of BCP3, BCP4, and MDC1. The coloring of amino acids is according to their polarity. (D) Sequence alignment of MDC1 and BCP4 point mutants comprising γH2A.X-binding site. MDC1 and BCP4 amino acids involved and predicted in binding of γH2A.X are indicated in red (top panel). Phosphopeptide pull-down analysis of wild-type and mutant GST-tBRCT domain of BCP4 (bottom panels). GST-tBRCT concentrations used with phosphopeptide were 2.5, and 20 mg/ml, respectively. Input lanes were loaded with 1.5 μg. Proteins were analyzed on 12% SDS-PAGE and gels were stained with Coomassie blue. Source data are available online for this figure.

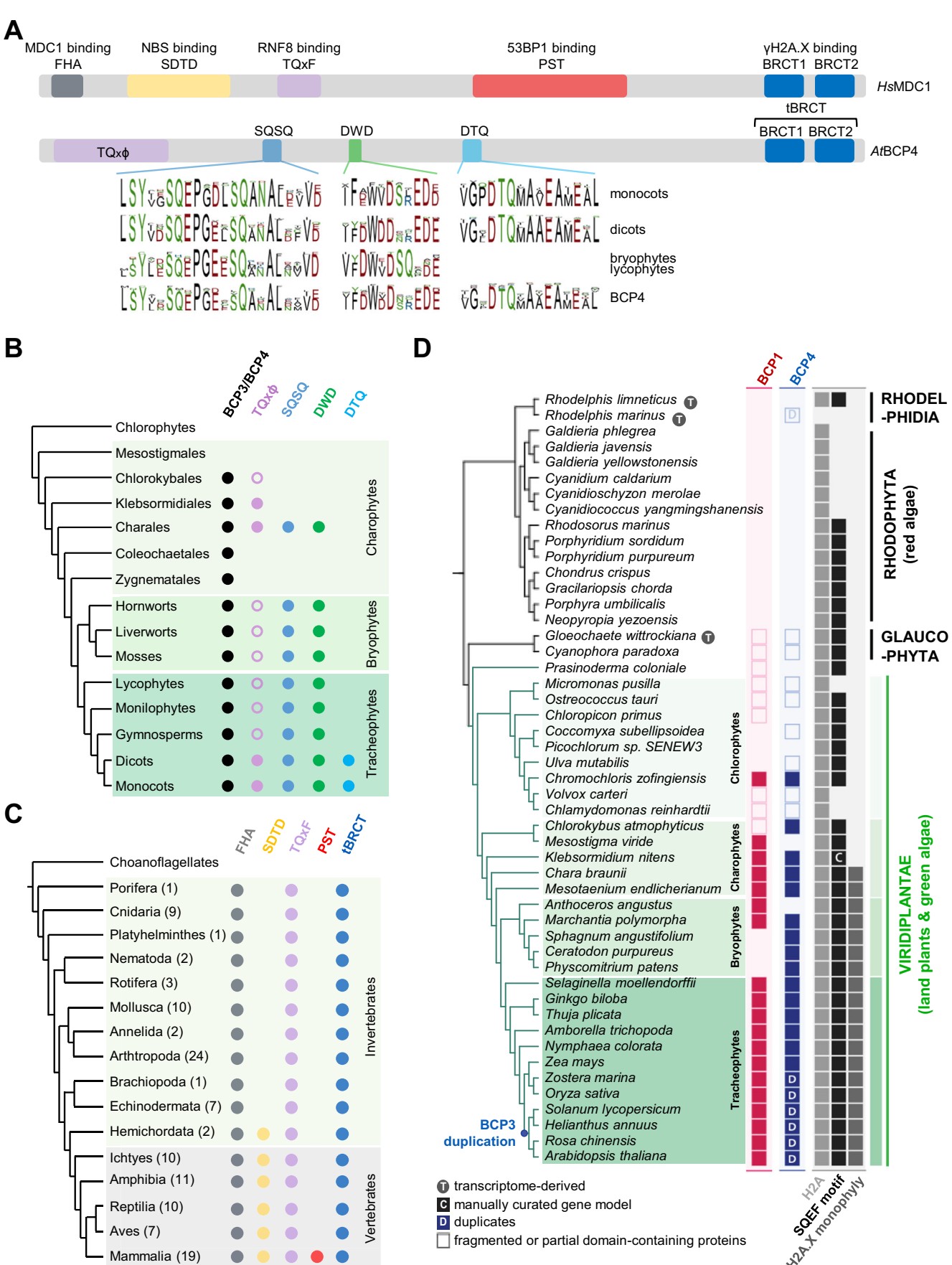

**Figure 5.   Comparison of BCP4 and MDC1 sequence motifs.**

(A) Schematic representation of human MDC1 and *Arabidopsis* BCP4 proteins. Conserved sequence motifs are indicated, and consensus sequences of plant motifs are depicted at the bottom. (B) Summary of the presence/absence of BCP3/BCP4 sequence motifs across Viridiplantae. Open circles denote phylogenetic groups where motif is present in only a subset of analyzed species. (C) In metazoa, functional motifs identified in human MDC1 are unique to mammals. The numbers of MDC1 proteins from each phylogenetic group used for the creation of cladogram are indicated in parentheses. (D) Evolutionary trajectories of BCP1, BCP4, and H2A.X in Archaeplastida. Protein presence is displayed at the tip of each branch, and major groups are denoted next to phylogeny. For H2A.X, (i) the presence of H2A, (ii) SQEF or SQEF-like motifs, and (iii) the presence of a monophyletic clade containing *Arabidopsis* H2A.X (Fig. EV4) were displayed separately. Data information: Plant and metazoan protein sequences used for the analysis shown in A–C are available in Source data. Source data are available online for this figure.

## BCP1 and BCP4 localize to DNA damage-induced foci in H2A.X-dependent manner but only BCP4 binds γH2A.X

Because BCP1 and BCP4 were most closely related to human PAXIP1 and MDC1 that bind γH2A.X (Fig. 1B–D), we put our focus on these two proteins. To test whether BCPs bind directly γH2A.X we performed two sets of experiments, co-immunostaining of mClover3-tagged BCPs and γH2A.X after induction of DSBs with bleomycin and in vitro phospho-H2A.X peptide-binding assays.

We used BCP1-mClover3 and BCP4-mClover3 complementing lines and performed co-immunostaining of mClover3 and γH2A.X in nuclei from seedlings after a 2-h treatment with bleomycin. Both BCP1 and BCP4 co-localized with γH2A.X in numerous foci (Fig. 3A). In *hta3 hta5* double-mutant plants, where H2A.X is not present (Lorković et al, 2017), BCP1-mClover3 (Fig. 3B) and BCP4-mClover3 (Fig. 3C) did not form foci after bleomycin treatment. Instead, both fusion proteins were diffusely distributed throughout the nucleoplasm and accumulated in nucleoli, demonstrating that the formation of BCP1 and BCP4 foci following DSBs induction depends on γH2A.X.

We expressed and purified both C- and N-terminally His$_6$-tagged BCP4 from insect cells and tested its capacity to bind the phosphorylated or unphosphorylated peptide comprising the 18 C-terminal amino acid residues of H2A.X. Affinity pull-down with unphosphorylated and phosphorylated H2A.X peptides revealed that BCP4 specifically binds phosphorylated H2A.X peptide (Fig. 3D). We also performed peptide-binding assays with the GST-tagged tBRCT domain of BCP4, which also specifically bound the phosphorylated H2A.X peptide (Fig. 3E). These results established that the C-terminal tBRCT domain of BCP4 directly interacts with the H2A.X phosphopeptide. In contrast to the tBRCT domain of BCP4, the purified GST-tagged N- and C-terminal tBRCT domains of BCP1 (tBRCT-N and tBRCT-C) did not interact with the phosphorylated H2A.X peptide (Fig. 3F), suggesting that these domains are not H2A.X phosphoserine binding modules.

Altogether, our results suggest that co-localization of BCP1 with γH2A.X foci is not due to direct γH2A.X binding but rather due to interaction between BCP1 and other DDR factor(s). These observations, along with similarities between the tBRCT domains of BCP1 and PAXIP1 (Fig. 1D), suggest that BCP1 may be functionally related to the metazoan PAXIP1, whose association with DNA damage foci depends on the presence of MDC1/RNF8 but not on direct interaction with γH2A.X (Gong et al, 2009; Muñoz et al, 2007). In contrast, based on the direct interaction of BCP4 (this work) and BCP3 (Fan et al, 2022) with γH2A.X and their co-localization with γH2A.X foci, we conclude that BCP3 and BCP4 have functional properties as human MDC1.

## BCP3 and BCP4 interaction with γH2A.X share properties with metazoan MDC1

Although amino acids residues of human MDC1 involved in binding phosphorylated serine of H2A.X are only partially conserved in *Arabidopsis* BCP4 (Fig. 1C) we showed that BCP4 binds γH2A.X and co-localizes with γH2A.X. We used AlphaFold models of tBRCT domains of BCP3 and BCP4 and superimposed them with a structure of human MDC1 in complex with phosphorylated H2A.X (PDB: 2AZM; Stucki et al, 2005). Despite the highly diverged primary sequence of human MDC1 and *Arabidopsis* BCP3 and BCP4 tBRCT domains with only ~24% identity (Fig. 1C), the three tBRCT domains displayed overall highly similar structures with root mean square deviations of <2 (Fig. EV3A,B). Like human MDC1, BCP4 and BCP3 were predicted to bind the phosphate moiety through direct interactions with side chain atoms of three structurally conserved residues: Asn864/608 (corresponding to MDC1 Lys1936), and Gln823/567 (corresponding to MDC1 Gly1899) and Thr862/606 (Fig. 4A). Gln823/567 of BCP4/BCP3 were also predicted to contact the H2A.X peptide main chain (Fig. 4A). In addition, interactions of MDC1 Arg1933 with the H2A.X C-terminus and with Glu2063 located on α3' (Stucki et al, 2005) are likewise conserved and are mediated by Arg861/605 and Glu997/740 of BCP4/BCP3 (Fig. 4B). Except for Gln823/567 the amino acids of the tBRCT domains predicted to contact H2A.X phosphoserine were conserved among plant BCP3 and BCP4 proteins (Fig. 4C). Analysis of the tBRCT domain of BCP1 and BCP2 revealed reduced abilities to form contacts with H2A.X peptide (Fig. EV3C–F), suggesting the lack of direct interaction with phosphorylated H2A.X.

Mutation of three predicted phosphoserine-contacting residues almost completely abolished phosphopeptide binding by the BCP4 tBRCT (Fig. 4D, BCP4M5), while single (Gln823Ala) and double (Asn864Ala and Thr862Ala) mutations of predicted phosphoserine-contacting residues had less prominent effects on binding (Fig. 4D, BCP4M3 and BCP4M4). Mutation of Arg1933 of human MDC1, which interacts with the H2A.X C-terminus and with Glu2063 located on α3', strongly affected interaction (Stucki et al, 2005). In BCP4, mutations of two arginine residues (Arg861Gln and Arg 863Gln) reduced binding (Fig. 4D, BCP4M2), although not to the extent observed with human MDC1 (Stucki et al, 2005). We also obtained BCP4 tBRCT domain with mutations mimicking human MDC1 phosphoserine-binding pocket and found similar binding efficiency as WT BCP4 (Fig. 4D, BCP4M1). We conclude that BCP3 and BCP4 bind phosphorylated H2A.X via conserved residues that form a pocket resembling that of human MDC1.

Although functional analyses and the phylogenetic position indicated that BCP3 and BCP4 and MDC1 are likely functional

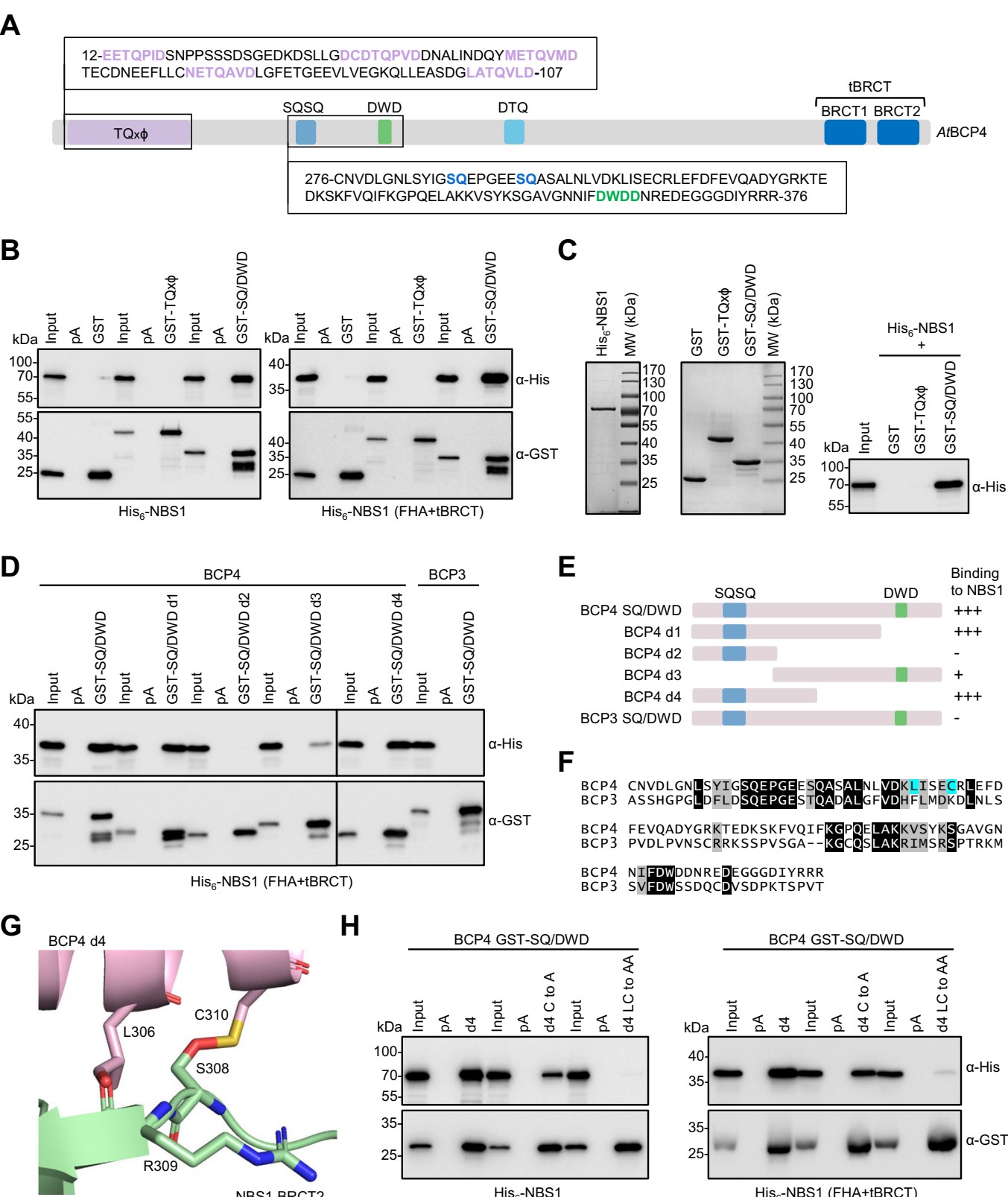

**Figure 6. BCP4 interacts with NBS1 of the MRN complex.**

(A) Schematic presentation of BCP4 with sequences of TQxϕ and SQ/DWD sequences used for protein expression. (B) Co-purification of full-length NBS1 and the FHA+tBRCT domains of NBS1 with GST and GST-tagged TQxϕ or SQ/DWD motifs of BCP4. (C) GST pull-down with independently purified recombinant proteins. Coomassie-stained gels of purified proteins are on two left panels. (D) Co-purification of His$_6$- FHA+tBRCT of NBS1 with the SQ/DWD deletion mutant forms of BCP4. (E) Schematic presentation of BCP4 SQ/DWD deletion mutant forms and summary of interactions with NBS1. (F) Sequence alignment of the SQ/DWD motifs of BCP3 and BCP4. Identical and similar amino acids are highlighted on black and gray background, respectively. Blue background indicates amino acids predicted to interact with NBS1. (G) AlphaFold2 predicted interaction of NBS1 and the SQ/DWD motif of BCP4. Amino acids predicted to interact are indicated with numbers corresponding to full-length proteins. (H) Co-purification of full-length NBS1 or the FHA+tBRCT domains of NBS1 with GST-tagged deletion 4 of BCP4 SQ/DWD motif and its point mutants as indicated. Source data are available online for this figure.

counterparts, BCP3 and BCP4 apparently lack the domains present N-terminal of the C-terminal tBRCT domain in MDC1 (Figs. 1A and 5A). We thus extended our survey of BCP3 and BCP4 orthologs across plants to detect additional conserved domains. The N-terminal intrinsically disordered regions of BCP3 and BCP4 did not display high sequence conservation among plant species. However, we identified three conserved regions (SQSQ, DWD, and DTQ) in most species analyzed (Figs. 5A,B and EV2C). Two of these motifs carry SQ and TQ dipeptides, which are potential targets of ATM/ATR kinases. However, the potential functional importance for these motifs remains to be determined. The DTQ motif was not present in BCP4 from non-flowering land plants and DWD motif was not present in multicellular algae (except *Chara*) (Fig. 5B). In addition, a careful inspection of BCP4 sequence alignments revealed a variable number (from three to ten, depending on the species) of TQxϕ (ϕ for hydrophobic amino acids) motifs, which resemble TQxF motifs found in human MDC1 (Fig. 5A). Thus, the TQxϕ motifs could be functional counterparts of TQxF motis binding RNF8 in human MDC1 (Kolas et al, 2007; Mailand et al, 2007). The TQxϕ motifs were absent in BCP4 from most multicellular green algae (except from *Chara* and *Klebsormidium*) (Fig. 5B, Charophytes) and were less prominent in BCPs from bryophytes, lycophytes, and gymnosperms (Fig. 5B, open circles). Along with the presence of TQxϕ motifs in BCP4 from basal dicots and their absence from BCP3 in dicots and monocots, these data suggested that BCP4 was ancestral and that the loss of TQxϕ motifs in BCP3 evolved recently after gene duplication in eudicots.

Despite sharing similar motifs and tBRCT domains with the human MDC1, BCP3, and BCP4 had neither the FHA domain nor the SDTD and PST repeats present in human MDC1 (Fig. 5A). The absence of SDTD and PST repeats in MUTATOR2, an MDC1 homolog from *Drosophila*, (Dronamraju and Mason, 2009; Kasravi et al, 1999), prompted us to re-examine the criteria that could define MDC1 in eukaryotes. We analyzed 119 metazoan MDC1 proteins, ranging from sponges to mammals, and found that all metazoan MDC1 proteins have an FHA domain at the N-terminus and tBRCT domain at the C-terminus (Figs. 5C and EV2D). A full repertoire of repeats, as in human MDC1 (TQxF, SDTD, and PST), was present only in mammals (Figs. 5C and EV2D). In other vertebrates, MDC1 homologs had TQxF and SDTD repeats. In invertebrates, MDC1 had only TQxF repeats, similar to the plant BCP4. Taken together, our analysis indicates that an ancestral MDC1 possessed a C-terminal tBRCT domain and that BCP3 and BCP4 can be considered as plant MDC1 homologs that acquired additional domains distinct to those acquired by MDC1 homologs in vertebrates.

To obtain more insights into the evolution of BCP1 and BCP3/BCP4 and their potential co-evolution with histone H2A variant carrying SQEF/Y motif in plants (referred to as H2A.X; Malik and Henikoff, 2003), we analyzed the presence of orthologs in representative taxa from the major clades of Archaeplastida. Histone H2A sequences possessing a SQEF/Y motif or SQEF-like (SQ + E/D + F/I/L/V/Y) motif at the C-terminus were present in most Archaeplastida lineages, except for some unicellular green algae (e.g., Chlamydomonadales), glaucophytes and extremophilic red algae (Fig. 5D). We were not able to unambiguously identify BCP1 and BCP4 candidate proteins in most unicellular green algae (chlorophytes) and glaucophytes (Fig. 5D, open boxes). Red algae lacked any sequences with the signatures of BCP1 and BCP4 or MDC1 from Opisthokonts, presumably because genome reduction contributed to massive gene losses in this group (Qiu et al, 2015). A clear phylogenetic distinction between the SQEF/Y motif-containing histone H2A and H2A.X, as defined by the presence of a monophyletic clade containing *Arabidopsis* H2A.X, appeared by the onset of more complex multicellular Streptophytes (Charales and Zygnematales) (Figs. 5D and EV4). This event occurred concurrently with the appearance of BCP1 and BCP4 (Fig. 5D), suggesting a degree of co-evolution with H2A.X.

## BCP4 interacts with NBS1 of the MRN complex

MDC1 not only targets sites of DNA damage through binding γH2A.X but also serves as a scaffold protein for the recruitment of various DNA damage response factors, including the MRN complex that interacts via NBS1 with SDTD motifs of human MDC1 (Chapman and Jackson, 2008; Melander et al, 2008; Spycher et al, 2008; Wu et al, 2008). Although the MRN complex is conserved in the plant lineage (Yoshiyama et al, 2013), the absence of the SDTD motifs in plants BCP3 and BCP4 made us to wonder if motifs we described in BCP3 and BCP4 were able to interact with the *Arabidopsis* NBS1 ortholog (Akutsu et al, 2007; Waterworth et al, 2007). We co-expressed GST-tagged TQxϕ and SQ/DWD motifs of BCP4 (Fig. 6A) along with GST alone with His-tagged full-length NBS1 and FHA+tBRCT of NBS1 in bacteria and analyzed co-purification of GST and His-tagged proteins on glutathione magnetic beads. Both the full-length and the FHA+tBRCT of NBS1 were co-purified only with SQ/DWD fragment of BCP4 (Fig. 6B). We also expressed and purified His-NBS1 and GST-TQxϕ and GST-SQ/DWD separately and tested them for interaction in vitro. Again, only GST-SQ/DWD pulled down His$_6$-NBS1, suggesting that *Arabidopsis* NBS1 interacted with the region of BCP4 comprising SQ and DWD motifs (Fig. 6C).

Deletion analysis of SQ/DWD fragment revealed that the DWD motif was dispensable for NBS1 interaction and that the interaction site very likely also did not involve the SQ motif (Fig. 6D,E). The SQ/DWD fragment of BCP3 did not interact with NBS1 (Fig. 6D,E), which could be explained by the sequence divergence between BCP3 and BCP4 (Fig. 6F). We used AlphaFold2 (Jumper et al, 2021, Evans et al, 2022) to predict interactions between the fragment of NBS1 containing FHA+tBRCT with shortened SQ/DWD region. AlphaFold2 predicted an interaction with an alpha helix region upstream of the SQ motifs and involving Leu306 and Cys310 of BCP4 and Ser308 and Arg309 of the second BRCT domain of NBS1 (Figs. 6G and EV5A,B). Mutations of Cys310 alone or Leu306 and Cys310 reduced or almost completely abolished interaction with NBS1 (Figs. 6H and EV5C). The Leu306 and Cys310 residues were not conserved in BCP3 SQ/DWD motif (Fig. 6F), potentially explaining the lack of in vitro interaction of BPC3 with NBS1. Altogether our results show that BPC4 binds γH2A.X and interacts specifically with NSB1 thus enabling the recruitment of the MRN complex, supporting further that BCP4 is a functional counterpart of mammalian MDC1.

# Discussion

In this study, we have surveyed the BRCT domain proteins in the plant lineage and characterized in detail the function of *Arabidopsis* proteins BCP1-4 with tBRCT domains. Of these four proteins only BCP4 protein revealed functional properties similar to that of metazoan MDC1. BCP4 binds to phosphorylated H2A.X in vitro, co-localizes with DNA damage-induced foci in vivo in an H2A.X-dependent manner, and binds NBS1 of the MRN complex. We thus conclude that BCP4 is the *Arabidopsis* functional counterpart of MDC1 and propose that it could be renamed AtMDC1.

Because most metazoans do not contain the domains found in mammalian orthologs of MDC1, a phylogenetic approach was necessary to reach a more accurate definition of the essential features that define MDC1 identity. This also showed that the presence of different functional motifs gradually increased in complexity during evolution of MDC1 in metazoans. Similarly, the complexity of conserved domains in BCP4 increased during plant evolution. The only common feature of MDC1 between Opisthokonts and plants is the presence of the C-terminal tBRCT domain that binds γH2A.X and the structural features of the molecular interactions involved in γH2A.X binding. We propose that these features should be used in future studies to find functional counterparts of MDC1.

In mammals, MDC1 acts as a scaffold for recruitment of DDR effectors through interactions with sequence motifs N-terminal of the tBRCT domain (Coster and Goldberg 2010). Although these motifs are absent in BCP4, other conserved domains evolved to mediate the recruitment of downstream effectors of DDR in plants. Of the three conserved regions (TQxϕ, SQSQ, and DTQ; see Fig. 5) that could potentially mediate such interactions we showed that the region downstream of the BCP4 SQSQ motif serves as an NBS1 interaction site. This interaction differs in two ways compared to that of human NBS1 and MDC1. First, unlike human NBS1 that interacts with SDTD motif in a phosphorylation-dependent manner (Chapman and Jackson, 2008; Melander et al, 2008; Spycher et al, 2008; Wu et al, 2008), *Arabidopsis* NBS1 binds a

region of BCP4 that does not require phosphorylation. Second, interaction of *Arabidopsis* NBS1 with BCP4 is predicted to engage only the BRCT2 domain of NBS1, whereas binding of human NBS1 to MDC1 includes both FHA and tBRCT domain (Lloyd et al, 2009; Williams et al, 2009; Hari et al, 2010). Although our data strongly support interaction of NBS1 and BCP4, we cannot exclude the possibilities (i) that the SQ dipeptides upstream of the NBS1 binding site are phosphorylated upon induction of the DNA damage and thereby modulate NBS1 binding as well as (ii) the existence of other interaction sites which would involve FHA domain of BCP4 as described for human NBS1 (Lloyd et al, 2009; Williams et al, 2009; Hari et al, 2010).

The roles of the conserved regions TQxϕ, SQSQ, and DTQ in BCP3 and BCP4 remains unclear, and it is possible that phosphorylation regulates their function in vivo, as documented for the conserved motifs SDTD and TQxF of human MDC1 (Coster and Goldberg, 2010). One possibility is that they serve as docking sites for other DDR effectors that may be shared with animals. We suggest that BCP1 is such an effector which, like human PAXIP1 (Gong et al, 2009), localized to DNA damage-induced foci in a γH2A.X interaction-independent manner (this work). In support to this idea, *bcp1* mutants showed reduced homologous recombination rates (Vladejić et al, 2022; Yu et al, 2023), in a manner comparable to PAXIP1-depleted chicken and HeLa cells (Wang et al, 2010). These data, together with sequence similarities in tBRCT domains (Fig. 1B–D), further support that BCP1 is a functional counterpart of PAXIP1 acting downstream of γH2A.X recognition by BCP4. Thus, DDR effector/adapter proteins very likely co-evolved with MDC1 independently in the major phylogenetic groups. However, whether BCP1 interacts with BCP4 remains to be determined.

In conclusion, we propose that plants do contain a functional counterpart of the mediator of DNA damage response (MDC1), which has evolved independently from Opisthokont MDC1 and acquired motifs distinct from those found in metazoan MDC1. Our work could serve as a framework that can be further expanded in the future by functional analyses of BCP4 motifs and identification of potential binding factors such as BCP1. It will also be important to monitor dynamics of BCP4, BCP1, NBS1 (MRN) foci formation in genetic backgrounds depleted of components of plant DDR. Our observations that red algae which are living in very harsh conditions (Cho et al, 2023), do not encode some core components of DDR raises the question of how these organisms deal with the DNA damage. More detailed proteome analysis combined with experimental approaches will be necessary to further explore this interesting question.

# Methods

## Identification of *Arabidopsis* BRCT proteome

We downloaded the full complement of human BRCT domain proteins (Woods et al, 2012) and performed BLAST searches on TAIR10 with each of them. All *Arabidopsis* hits were then analyzed on ScanProsite and InterProScan to identify BRCT and other annotated domains. In the next step, we used BRCT domains from identified proteins for BLAST searches on TAIR10 with iterations with new BRCT hits until no new ones were detected. All new hits

**Table 1. Oligonucleotides used for genotyping and cloning.**

**Oligonucleotides used for genotyping**

| | |
|---|---|
| BCP1-1-LP | AGATTTGAATGGGATTCCAGG |
| BCP1-1-RP | CCAAAGTATCAGTCTCTGGCG |
| BCP1-2-LP | GATGGTCTTTCTCTTCTGGGG |
| BCP1-2-RP | CGCCAGAGACTGATACTTTGG |
| BCP4-1-LP | ACGAACATGGAGTTTCTGGTG |
| BCP4-1-RP | CTTTTACTTGCAACGCCAAAG |
| BCP4-2-LP | CTGCCTTGCATTCTTTTCAAG |
| BCP4-2-RP | TGTAAGACAACTCGCCTCACC |
| BCP3-LP | CACGCATCAAATCTAGCCAAG |
| BCP3-RP | ATCTTCAATTTCCCCACATCC |
| BCP2-1-LP | TTGTTGGGCAGACAAAGAATC |
| BCP2-1-RP | GAGTTTTCCTGACTTTTCCGG |
| BCP2-2-LP | GAGTTTTCCTGACTTTTCCGG |
| BCP2-2-RP | TTGTTGGGCAGACAAAGAATC |
| H2A.X.3-LP | ATCACTCCACTCACAAAATCCTC |
| H2A.X.3-RP | TGGAACAGAGAGCCATGTCTATG |
| H2A.X.5-LP | CCTAAAGCCCACTCATCTTCTC |
| H2A.X.5-RP | CGAATCCAAACAAGAGAACTGAAC |

**Oligonucleotides used for cloning**

| | |
|---|---|
| BCP1Nde-pET28a For | ATCATCATATGCAATCGGATTCGGGTTTGCC |
| BCP1Sal-pET28a Rev | GCTCAGTCGACTTAATGGTACACACACAAATC |
| BCP4Nde-pET15b For | ACCTCCATATGGCTAAATCTAACCAAAACTT |
| BCP4Bam-pET15b Rev | GCTCTGGATCCTTACCCGCTACGACGTTGGA |
| BCP3Nde-pET28a For | ATCATCATATGGAAACCGAAGATTTCGCCTC |
| BCP3Sal-pET28a Rev | GTTGCGTCGACTTATAATCTCTGATTTTGGT |
| BCP1tBRCT1BamHI-pGEX4T1 For | CTAATGGATCCTTGCCTCCCAAGACGTATTCG |
| BCP1tBRCTSal-pGEX4T1 Rev | CTAACGTCGACCTACTCGTAATCAACCTCAGGTAG |
| BCP1tBRCT2BglII-pGEX4T1 For | CTAATAGATCTGTCTTCCAGGACCAAGAACATG |
| BCP1tBRCT2Sall-pGEX4T1 Rev | CTAATGTCGACCTAACAAACGTACTCCACCAGGTAATCG |
| BCP3tBRCTBamHI-pGEX4T1 For | CTAATGGATCCGGTAAAATAGGTGACTTCGTG |
| BCP3tBRCTSal-pGEX4T1 Rev | CTAACGTCGACTTATAATCTCTGATTTTGGTG |
| BCP4tBRCTBamHI-pGEX4T1 For | CTAATGGATCCATCTCCGAGACAAAGAGTACTAG |
| BCP4tBRCTSal-pGEX4T1 Rev | CTAACGTCGACTTACCCGCTACGACGTTGGAAC |
| NBS1Nhe-pET28a For | CTAGGGCTAGCGTTTGGGGTCTCTTTCCCGTTGATC |
| NBS1Sall-pET28a Rev | ATCGAGTCGACTCAACTTCCAGAGAGAAACCCGCG |
| NBS1(F+tBRCT)Nde-pET28a For | CTAGCATATGGTTTGGGGTCTCTTTCCCGTTGATC |
| NBS1(F+tBRCT)Sall-pET28a Rev | ATCGAGTCGACTCATAGATTTCCGGAAAAGACGGCG |
| BCP4TQEcoRI-pGEX4T1 For | ATCTAGAATTCGAAGAGACCCAGCCGATTG |
| BCP4TQxSal-pGEX4T1 Rev | ATCTAGTCGACCTAATCCAAAACCTGAGTAGCC |
| BCP4SQBam-pGEX4T1 For1 | CTAATGGATCCTGCAATGTCGATCTGGGGAATTTG |
| BCP4SQBam-pGEX4T1 For1 | CTAATGGATCCGAGTGCCGTCTAGAGTTTGAT |
| BCP4SQSal-pGEX4T1 Rev1 | CTAACGTCGACTTATCTTCTACGATAGATATCACC |
| BCP4SQSal-pGEX4T1 Rev2 | CTAACGTCGACTTAAACAGCTCCACTTTTGTAACT |
| BCP4SQSal-pGEX4T1 Rev3 | CTAACGTCGACTTACTCACTAATGAGCTTATCAAC |

**Table 1.** (continued)

| BCP4SQSal-pGEX4T1 Rev3 | TAACGTCGACTTATTTCCTTCCATAGTCAGCTTG |
|---|---|
| BCP3SQBam-pGEX4T1 For | CTAAT**GGATCC**GCTAGTAGCCATGGTCCAGGGC |
| BCP3SQSal-pGEX4T1 Rev | TAACGTCGACTTAAGGGTCACTCACATCACACTG |

were collected and analyzed on ScanProsite, InterProScan, and Uniprot to identify BRCT and other conserved domains.

To identify homologs of BCP1-4 in other plant species, *Arabidopsis* BRCT domain protein sequences were used in BLAST searches of Phytozome (https://phytozome.jgi.doe.gov/pz/portal.html), Fernbase (https://www.fernbase.org/), ORCAE (https://bioinformatics.psb.ugent.be/orcae/overview/Chbra), water-lilyPond (http://waterlily.eplant.org/) and MarpolBase (https://marchantia.info/) websites. All sequences were aligned with CLC Genomics Workbench 11.0, and sequences with long insertions or deletions were removed before performing final alignments and phylogenetic analysis with CLC Genomics Workbench 11.0.

Metazoan MDC1 proteins were identified by NCBI BLAST searches and manually inspected for the presence of conserved motifs present in human MDC1.

## Evolutionary reconstruction of proteins in Archaeplastida using clustering-based protein identification and phylogenetic analysis

To reconstruct the evolutionary history of H2A.X and BCP proteins in plant, we constructed orthologous gene clusters (i.e., orthogroups) in 49 representative proteome datasets from Archaeplastida comprising 46 genomes and three transcriptomes from two Rhodelphidia and one Glaucophyta (Source data file 5). Orthofinder v2.5.2 (Emms and Kelly, 2019) was used to cluster genes in a non-biased way by comparing each gene to the entire proteome dataset (>0.9 million proteins). After testing different parameters, we chose DIAMOND (Buchfink et al, 2015) or homology search with "-S diamond_ultra_sens" and adjusted inflation parameter "-I 2" based on previously validated BRCT proteome. The absence and presence of proteins (or domains) after manual correction of proteins were visualized next to the species tree using iToL v6.7 (Letunić and Bork, 2021). The internal relationships within the class level in the species tree were modified based on the Orthofinder output species tree, and higher taxonomic relationships (class or higher ranks) were verified based on currently accepted phylogenies (Leebens-Mack et al, 2019). MAFFT v7.310 (Katoh and Standley, 2013) was used to align protein sequences based on orthogroups. Individual maximum likelihood gene trees were built with IQ-TREE v2.1.2 (Minh et al, 2020), which used model selection ("-m TEST") and an ultrafast bootstrap approximation approach (1000 bootstrap replicates; "-bb 1000").

## Plant material and growth conditions

All *A. thaliana* plants used in this study are from the Colombia ecotype (Col-0). Single T-DNA insertion mutant BCP1-4 lines were obtained from the Nottingham Arabidopsis Stock Centre. *bcp1-1* (SALK_022790), *bcp1-2* (SALK_001578), *bcp4-1* (SAIL_1222_D03), *bcp4-2* (SALK_038422) *bcp3-1* (SALK_111173) *bcp2-1* (SALK_025100.24.70.x) and *bcp2-2* (SAIL_13_D01) homozygous

T-DNA insertions were verified by PCR genotyping (Table 1). Mutant lines *hta7* (GK_149G05); *hta3* (SALK_012255), and *hta5* (SAIL_382_B11) were previously described (Lorković et al, 2017; Yelagandula et al, 2014).

Plants for genotyping or generating transgenic lines were grown in fully automated climate chambers under long-day conditions (16 h light, 8 h dark). Plants used for DNA damage sensitivity assays, selection of transgenic lines, H2A.X phosphorylation assays following DNA damage treatment, and immunofluorescence analyses were grown on MS plates under sterile, long-day conditions (light intensity: 50 μM/m²/s).

## Analysis of DNA damage sensitivity

To assess sensitivity to DNA damage of mutants *bcp1, bcp2, bcp3,* and *bcp4*, sterilized seeds (64 seeds per replicate) were germinated on MS plates containing 20 μg/ml zeocin (Invitrogen). True leaf development was scored 14 days after germination with replicate numbers shown on each panel (Rosa and Mittelsten Scheid, 2014; Lorković et al, 2017).

## Generation of transgenic lines expressing BCP1, BCP3, and BCP4-mClover3 fusion proteins

For complementing *Arabidopsis bcp* mutant lines, DNA fragments of *BCP* genes (full genomic sequence) with the respective endogenous promoter (~1000 nucleotides upstream of start codon) were fused to the N-terminus of fluorescent protein-tag (mClover3) into the T-DNA binary vector pCBK02 (with either BASTA or spectinomycin selection marker) by using the Gibson assembly method. Plasmids were transformed into *A. tumefaciens* strain GV3101 and *Arabidopsis* plants transformed by floral dip method. Seeds from T3 transgenic lines, which were confirmed to be homozygous, were evaluated for complementation by true leaf assay and were also used for immunostaining.

## Analysis of H2A.X phosphorylation in *bcp* mutant lines

In total, 300 mg of 12–14 days old WT and *bcp1-4* mutant seedlings grown on vertical MS plates were transferred into liquid MS media in the presence or absence of 20 μg/ml bleomycin (Calbiochem). Following vacuum infiltration for 2 min, seedlings were incubated for 2 h under light on a shaker. After removal of excess medium, seedlings were frozen in liquid nitrogen and stored at −70 °C until further use. Nuclei were isolated as described in Lorkovic et al, (2017) and stored at −20 °C. Western blots were performed according to standard procedures using a γH2A.X antibody (Sigma-Aldrich, H5912). H2A.X (Yelagandula et al, 2014) antibody served as loading control for normalization. Primary antibodies were used at 1:1000 dilution and secondary goat anti-rabbit IgG coupled to HRP at 1:10,000 dilution.

Quantification of western blots was done with the ChemiDoc software (Bio-Rad) by using the volume tool and normalization to levels of unmodified H2A.X.

## Plasmids for expression of proteins in bacteria

Full-length *BCP1*, *BCP3*, *BCP4, and NBS1* cDNAs were amplified by RT-PCR from RNA isolated from WT seedlings treated with 20 µg/ml bleomycin for 2 h. RNA was isolated using the Spectrum Plant Total RNA-Kit (Sigma-Aldrich), following the manufacturer's protocol. Reverse transcription was done with the RevertAid H Minus First Strand cDNA Synthesis Kit (Thermo Fisher Scientific), following the manufacturer's protocol. For PCR amplification, 1 µl of the RT reaction was used with gene-specific primers for cloning. *BCP1* was cut with *Nde*I/*Sal*I and cloned into pET28a (Novagen), *BCP3* was cut with *Nde*I/*Sal*I and cloned into pET28a (Novagen)*, BCP4* was cut with *Nde*I/*Bam*HI and cloned into pET15b (Novagen). tBRCT domains of BCP1, BCP3, and BCP4 were amplified from the cDNA clones above and cloned into *Bam*HI/*Sal*I (BCP1 tBRCT-N, BCP3 tBRCT, and BCP4 tBRCT) and *Bgl*II/*Sal*I (BCP1 tBRCT-C) of pGEX-4T-1 (Cytiva). Full-length NBS1 and a fragment encoding FHA+tBRCT domains were cloned into pET28a as *Nhe*I/*Sal*I and *Nde*I/*Sal*I, respectively. The TQxϕ and SQ/DWD regions of BCP4 and SQ/DWD of BCP3 were cloned into pGEX-4T-1 as *Bam*HI/*Sal*I. Fragments encoding mutated BCP4 tBRCT and SQ/DWD domains were synthesized by Integrated DNA Technology (IDT) and cloned into pGEX-4T-1 as corresponding wild-type fragments. All primers used for cloning are listed in Table 1.

## Expression and purification of BCP1 and BCP4 from insect cells

The VBCF ProTech Facility (VBCF; https://www.viennabiocenter.org/vbcf/protein-technologies/) used the BCP1 and BCP4 pET expression plasmids described above to generate His$_6$-tagged versions (both N-terminal and C-terminal) in a baculovirus expression system. After selecting proper expression conditions, two liters of cells were collected and stored at −70 °C until further use.

Cells were thawed and resuspended in 20 ml lysis buffer (50 mM HEPES pH 7.5, 500 mM NaCl, 20 mM imidazole, 2 µl/ml Benzonase) containing protease inhibitors (Roche) and sonicated for 2 min (5"on/5"off, 40% amplitude). After centrifugation for 30 min at 20,500 × g, the supernatant was transferred into new tube, and 750 µl of Ni-NTA agarose (Qiagen), washed with lysis buffer, were added, and incubated for 1 h at 4 °C on a rotating wheel. Beads with bound proteins were collected by short centrifugation and transferred into a disposable gravity column. The column was washed with 5 ml of lysis buffer, followed by five washes with 5 ml of wash buffer (50 mM HEPES-KOH pH 7.5, 500 mM NaCl, 40 mM imidazole). Proteins were eluted in 300 µl fractions with elution buffer (50 mM HEPES pH 7.5, 500 mM NaCl, 500 mM imidazole). Fractions were analyzed with SDS-PAGE, and fractions containing eluted protein were pooled, concentrated to 500 µl, and stored at −20 °C until further use.

Affinity-purified samples were further purified by size exclusion chromatography over a Superdex 200 10/300 GL column (GE Healthcare) with running buffer (50 mM Tris-HCl pH 7.5, 500 mM NaCl, 0.05% NP-40, 1 mM DTT) using an NGC–MPLC System (Bio-Rad) and analyzed with the Image Lab Software (Bio-Rad).

Peak fractions were analyzed by SDS-PAGE, and those displaying desired purity of the protein were stored at −70 °C.

## Expression and purification of recombinant proteins expressed in bacteria

BL21 (DE3) RIL *E. coli* cells transformed with plasmids for protein expression were grown at 37 °C overnight in 200 ml LB. Cultures were diluted in 2 L of LB and grown for 3 h at RT and then induced for 5–7 h at RT with 1 mM IPTG. For GST-tagged proteins cells were collected and resuspended in 20 ml of extraction buffer (50 mM Tris-HCl pH 8.0, 1 M NaCl, 1 mM DTT, 0.1% Triton X-100) containing protease inhibitors (Roche), 10 µl of benzonase (1 mg/ml) and 50 mg of lysozyme. After sonication for 10 min at high intensity (15"on/15"off) and 5 min at medium intensity (15"on/15"off) extracts were centrifuged for 15 min at 4 °C at 40,000 × g. Extracts were incubated with 300 µl of glutathione Sepharose 4 fast flow (Cytiva) at RT for 1 h and then transferred to disposable columns and washed with 5 column volumes of extraction buffer. Proteins were eluted with six 300-µl elution steps with 50 mM Tris-HCl pH 8.0, 500 mM NaCl buffer containing 20 mM reduced glutathione and 1 mM DTT. Fractions were analyzed on 10–12% SDS-PAGE, pooled, and buffer was exchanged into 50 mM Tris-HCl pH 8.0, 0.5 M NaCl, 1 mM DTT.

For purification of His$_6$-tagged NBS1 cells were resuspended in extraction buffer (50 mM Tris-HCl pH 7.5, 500 mM NaCl, 2 mM DTT, 0.05% NP-40) and processed as for GST-tagged proteins. Extracts were incubated with 400 µl of Ni-NTA beads (Qiagen) for 1 h at RT and then transferred to disposable columns and washed with 5 column volumes of extraction buffer containing 5 mM imidazole. Proteins were eluted with six 300 µl elution steps with 50 mM Tris-HCl pH 7.5 buffer, containing 500 mM NaCl, 2 mM DTT, and 300 mM imidazole.

In all purifications, elution fractions were analyzed on 10–12% SDS-PAGE, pooled, and buffer was exchanged into 50 mM Tris-HCl pH 7.5, 150 mM NaCl, 1 mM DTT.

## Interaction of BCP4 TQxϕ and SQ/DWD motifs with NBS1

Plasmids expressing His$_6$-tagged full-length or FHA+tBRCT domains of NBS1 were co-transformed with plasmids expressing GST-tagged TQxϕ, SQ/DWD motifs or deletions and point mutants of SQ/DWD motif into *E. coli* BL21 (DE3) RIL. Ten milliliter cultures grown overnight at 37 °C were diluted into 200 ml of fresh LB medium and grown at RT for 3 h and then induced for 5 h at RT. Cells equivalent to 100 of culture were resuspended in 5 ml of TBS containing 0.1% Triton X-100, 1 mM DTT, protease inhibitors (Roche), 10 µl of benzonase (2 µg/ml) and lysozyme (50 mg per 50 ml). After sonication for 10 min at high intensity (10"on/15"off) and 5 min at medium intensity (10"on/15"off) extracts were centrifuged for 20 min at 4 °C at 40,000×g. Extracts were incubated with 50 µl of magnetic Glutathione (Thermo Fisher Scientific) or Protein A (Cytiva) beads (each with 2.5 ml of protein extract) at RT for 1 h. Beads were collected and washed seven times with extraction buffer (without benzonase and lysozyme) and finally resuspended in 80 µl of 1× SDS-PAGE loading buffer prepared in extraction buffer. Samples were run on 10–12% SDS-PAGE and analyzed by western blotting with anti-His (Sigma-

Aldrich, H1029) and anti-GST (Santa Cruz Biotechnology, sc-138) antibodies. All co-purification assays were done in duplicates.

We also performed interaction assays with independently purified His$_6$-NBS1 and GST-TQx and GST-SQ/DWD. GST-tagged proteins (5 μg) along with GST alone were incubated with 10 μl of magnetic glutathione beads (Thermo Fisher Scientific) for 1 h at RT, washed with interaction buffer (20 mM Tris-HCl, pH 7.5, 150 mM NaCl, 1 mM DTT, 0.1% Triton X-100) and then incubated with 5 μg of His$_6$-NBS1 for 90 min at RT. Beads were washed 6 times with binding buffer, denatured in 70 μl of 1× SDS-PAGE loading buffer and 10 μl were loaded on 10% SDS-PAGE and analyzed by western blotting with anti-His (Sigma-Aldrich, H1029) antibody. Input lanes were loaded with 1/20 of protein used for pull-down.

### Interaction of BCP4 with phosphorylated H2A.X

Peptide-binding assays were performed as described by Stucki et al (2005). Biotinylated peptides (25 μg) corresponding to the C-terminal tail of H2A.X, in phosphorylated and unphosphorylated form (biotin-PSKVGKNKGDIGSASQEF-OH and biotin-PSKVGKNKGDIGSASp-QEF-OH), were bound to 10 μl of Dynabeads MyOne Streptavidin T1 (Invitrogen) in 500 μl of binding buffer (50 mM Tris-HCl pH 7.5, 150 mM NaCl, 0.05% NP-40) for 30 min at room temperature. After washing the beads with binding buffer, purified proteins were diluted with binding buffer to obtain a buffer with a final concentration of 150 mM NaCl and mixed with streptavidin beads. From this step on, the binding buffer contained PhosStop (Roche) to prevent unwanted dephosphorylation of the peptides. Reactions were incubated on a rotator at 4 °C for 2 h. After three washes with 500 μl of binding buffer, beads were resuspended in 25 μl of 1× SDS-PAGE loading buffer, denatured, and 10 μl were run on 10% (full-length His$_6$-BCP4) or 12% (GST-tagged tBRCT domains) SDS-PAGE. Gels were stained with Coomassie blue, and images were acquired with the ChemiDoc imaging system (Bio-Rad).

### AlphaFold predictions

We ran AlphaFold-Multimer v2.2 (Evans et al, 2022) for all protein complex predictions. Of the five models generated, we used only the model ranked_3 for further processing and interpretation. We used the software PyMOL (TM) Molecular Graphics System, Version 2.5.0. Copyright (c) Schrodinger, LLC., for the visualization and super-imposition of AlphaFold models and X-ray structural data.

### Immunofluorescence on isolated nuclei

Approximately 15 seedlings were incubated in MS media either with or without bleomycin (20 μg/ml) for 2 h. After washing with MS medium, seedlings were fixed in Tris buffer (10 mM Tris-HCl pH 7.5, 10 mM EDTA, 100 mM NaCl) containing 4% formaldehyde for 20 min. Samples were washed once with Tris buffer and once with LB01 buffer (15 mM Tris-HCl pH 7.5, 2 mM EDTA, 0.5 mM spermine, 80 mM KCl, 20 mM NaCl, 0.1% Triton X-100). Fixed material was homogenized to a fine suspension by chopping with razor blades in a petri dish in 400 μl of LB01 buffer and filtered through a 40 μm cell strainer into an Eppendorf tube. The suspension was washed with 400 μl of LB01 buffer in the Petri dish, pipetted through the cell strainer into the same Eppendorf tube, and finally the mesh was also washed with 400 μl of LB01 buffer. Samples were spun

for 2 min at $2000 \times g$ at 4 °C, the supernatant was discarded, and the pellet was resuspended in 500 μl of LB01 buffer and kept on ice for ~10 min. This step was repeated until the pellet was no longer green. Finally, nuclei were pelleted for 2 min at $1000 \times g$ at 4 °C and resuspended in 100 μl of LB01 buffer. Of this suspension, 10 μl were transferred to a microscopic slide and dried completely at room temperature. Dried nuclei were fixed in 4% formaldehyde in PBS at room temperature for 30 min followed by two 5 min washes in PBS. Nuclei were blocked in 1% bovine serum albumin in PBS in a moist chamber at 37 °C for 30 min followed by one 5 min wash in PBS. Samples were incubated with primary antibodies diluted 1:100 (rabbit pAb α-γH2A.X from Sigma-Aldrich, H5912 and mouse mAb α-GFP from Roche, 11814460001) in 1% BSA (in PBS) and incubated in a moist chamber at 37 °C for 2 h. After three washes in PBS for 10 min, samples were incubated with Alexa flour labeled secondary antibodies (Invitrogen) diluted 1:200 in 1% BSA (in PBS) in a moist chamber at 37 °C for 30 min followed by three washes in PBS for 5 min. Slides were mounted in Vectashield (Vector laboratories) containing DAPI (1 μg/ml), sealed, and stored at 4 °C.

Microscopy was performed at the IMP/IMBA/GMI BioOptics facility using a LSM laser scanning confocal microscope (LSM720 Axio Observer, Zeiss). Images were analyzed with ZEN software (Zeiss) and γH2A.X foci were quantified with ImageJ.

## Data availability

This study includes no data deposited in external repositories.

## Peer review information

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

## Acknowledgements

The authors thank members of the Berger lab for their comments and helpful discussion. We thank Dr. Nicholas Irwin for sharing pipelines used for the evolutionary reconstruction of *BCP* genes. Authors acknowledge support from PlantS and ProTech facilities at the Vienna BioCenter Core Facilities (VBCF), and the BioOptics facility and Molecular Biology Services from IMP/IMBA/GMI, and Dr. J. Matthew Watson for proofreading the manuscript. This work was funded by core funding provided by the Gregor Mendel Institute. CHC was funded by the Basic Science Research Program through the National Research Foundation of Korea (RS-2023-00248097). FB received funds from the FWF grant PAT 1104523.

## Author contributions

**Zdravko J Lorković**: Conceptualization; Formal analysis; Supervision; Validation; Investigation; Visualization; Methodology; Writing—original draft; Writing—review and editing. **Michael Klingenbrunner**: Conceptualization; Formal analysis; Supervision; Validation; Investigation; Methodology; Writing—original draft; Writing—review and editing. **Chung Hyun Cho**: Formal analysis; Investigation; Methodology. **Frédéric Berger**: Conceptualization; Supervision; Funding acquisition; Writing—original draft; Project administration; Writing—review and editing.

## Disclosure and competing interests statement

The authors declare no competing interests.

# Expanded View Figures

**Figure EV1.   Phylogeny of BCP1, BCP3, and BCP4.**

(A) Maximum likelihood tree of BCP1 across Viridiplantae. The schematic presentation of BCP1 is shown at the bottom and a PHD finger present in all BCP sequences except in *Brassicaceae* is indicated. (B) Maximum likelihood tree of BCP3 and BCP4 across Viridiplantae. (A, B) Major clades are indicated by differently colored shading. The non-flowering land plant clade includes sequences from hornworts, mosses, liverworts, lycophytes, and monilophytes. *Thuja plicata* was the only gymnosperm used in the analysis. (C) Alignment of BCP3 and BCP4 proteins. Identical and conserved amino acids are indicated in red and blue letters, respectively. The positions of introns are indicated by black arrows. The tBRCT domain and three other conserved regions are shown in colored boxes. (D) Schematic representation of *Arabidopsis BCP* genes with exons indicated by gray boxes and introns by black lines. Exons and introns are drawn to scale according to the lengths of DNA sequences. Positions of T-DNA insertions in *bcp* mutant lines used for DNA damage sensitivity assays are indicated above each gene. Data information: A list of protein sequences used for the analysis is available from Source data for Fig. 5A,B.

▶

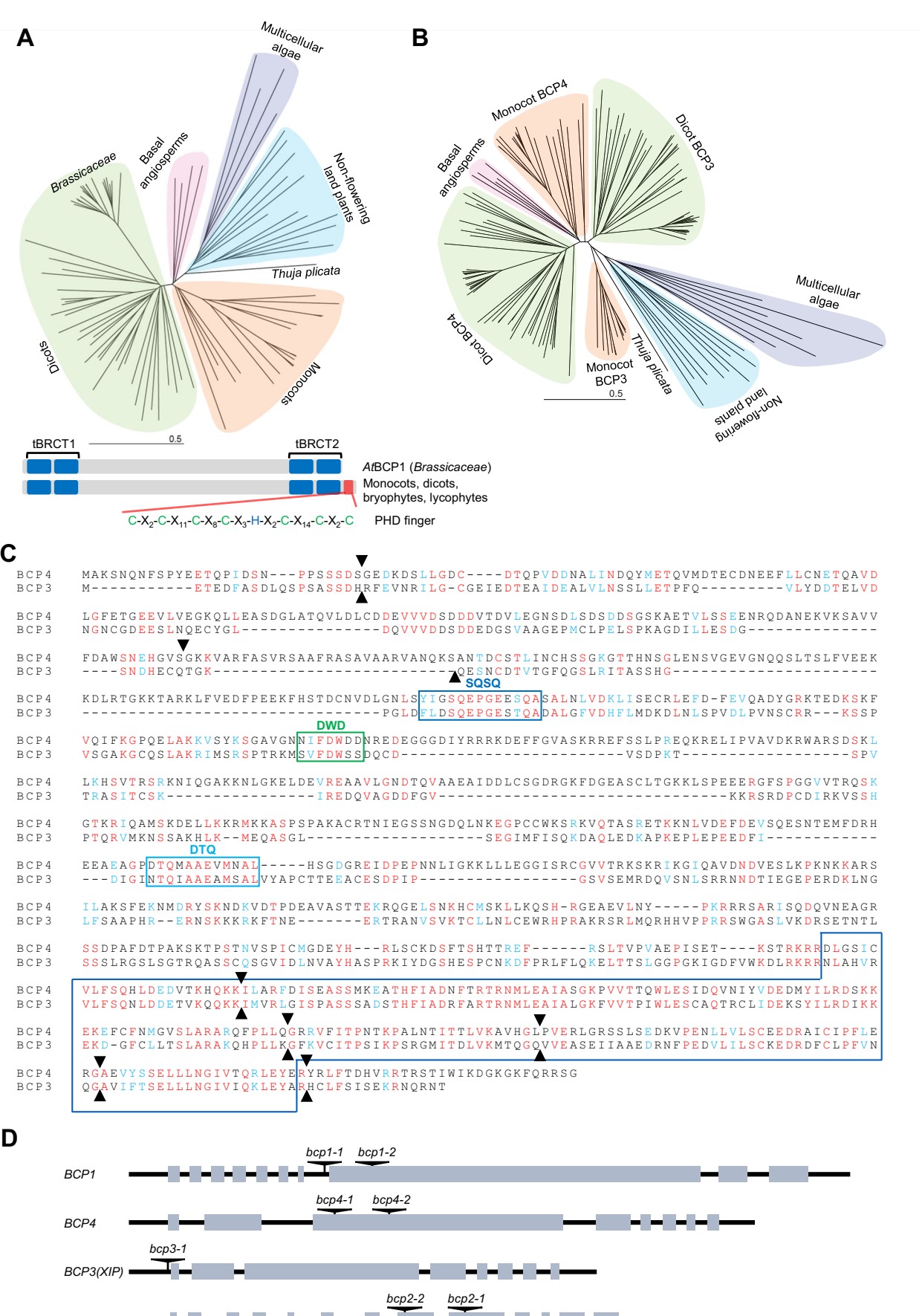

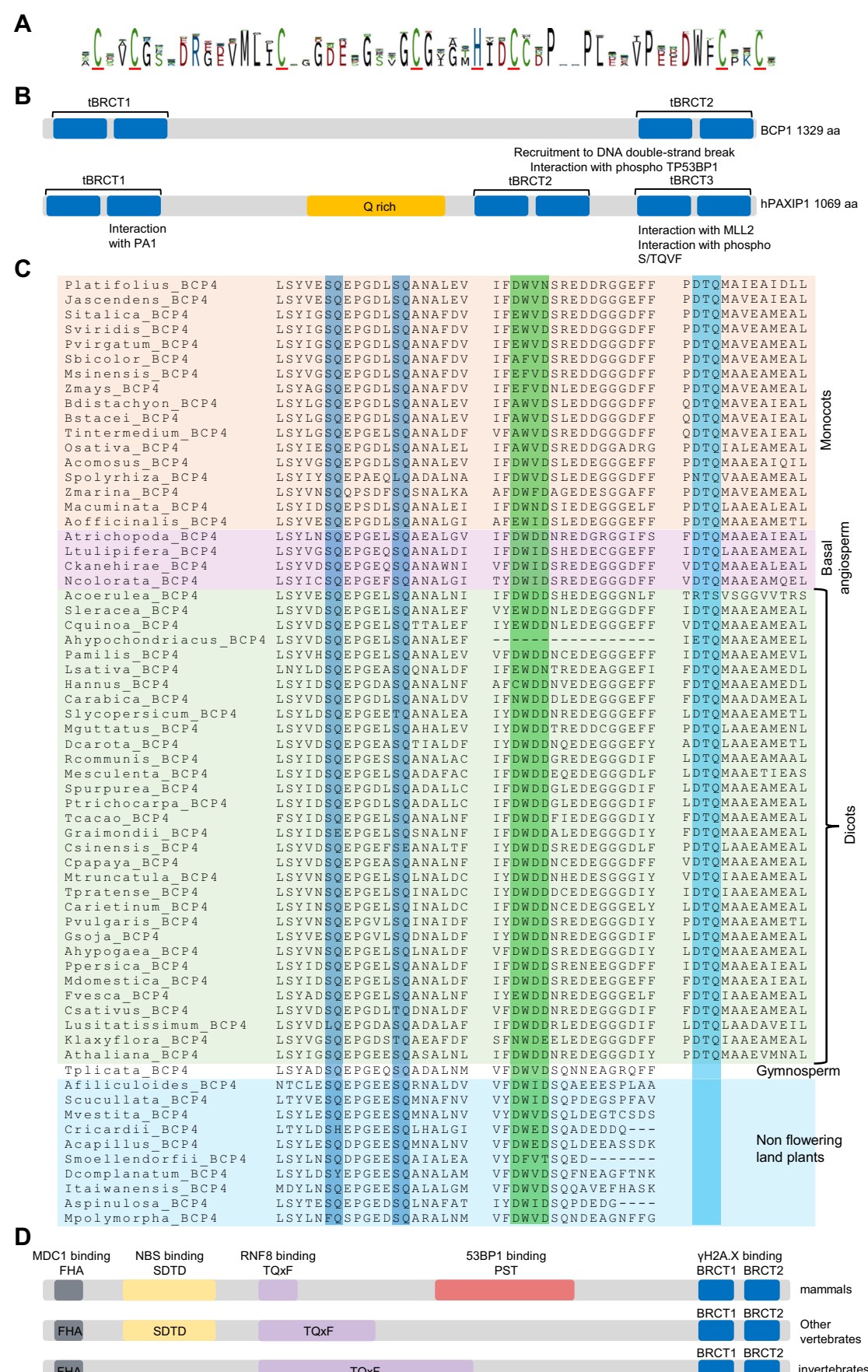

**Figure EV2.  Primary sequences analysis of BCP proteins.**

(A) The consensus sequence of the PHD finger from BCP1 derived from the alignment of plant BCP1 proteins. Cysteine and histidine residues characteristic of PHD fingers are underlined. (B) Schematic representation of *Arabidopsis* BCP1 and human PAXIP1 proteins. Conserved domains and motifs of PAXIP1 and their assigned functions are indicated. A list of plant species and the corresponding protein sequences used for the analysis in (A, B) are available from Source data. (C) Alignment of SQSQ, DWD, and DTQ sequence motifs from BCP4. The signature motifs are shaded in blue and green. (D) Schematic representation of MDC1 proteins from invertebrate, vertebrate (except mammals), and mammalian species. Conserved domains and motifs and their assigned functions are indicated. Source data are available online for this figure.

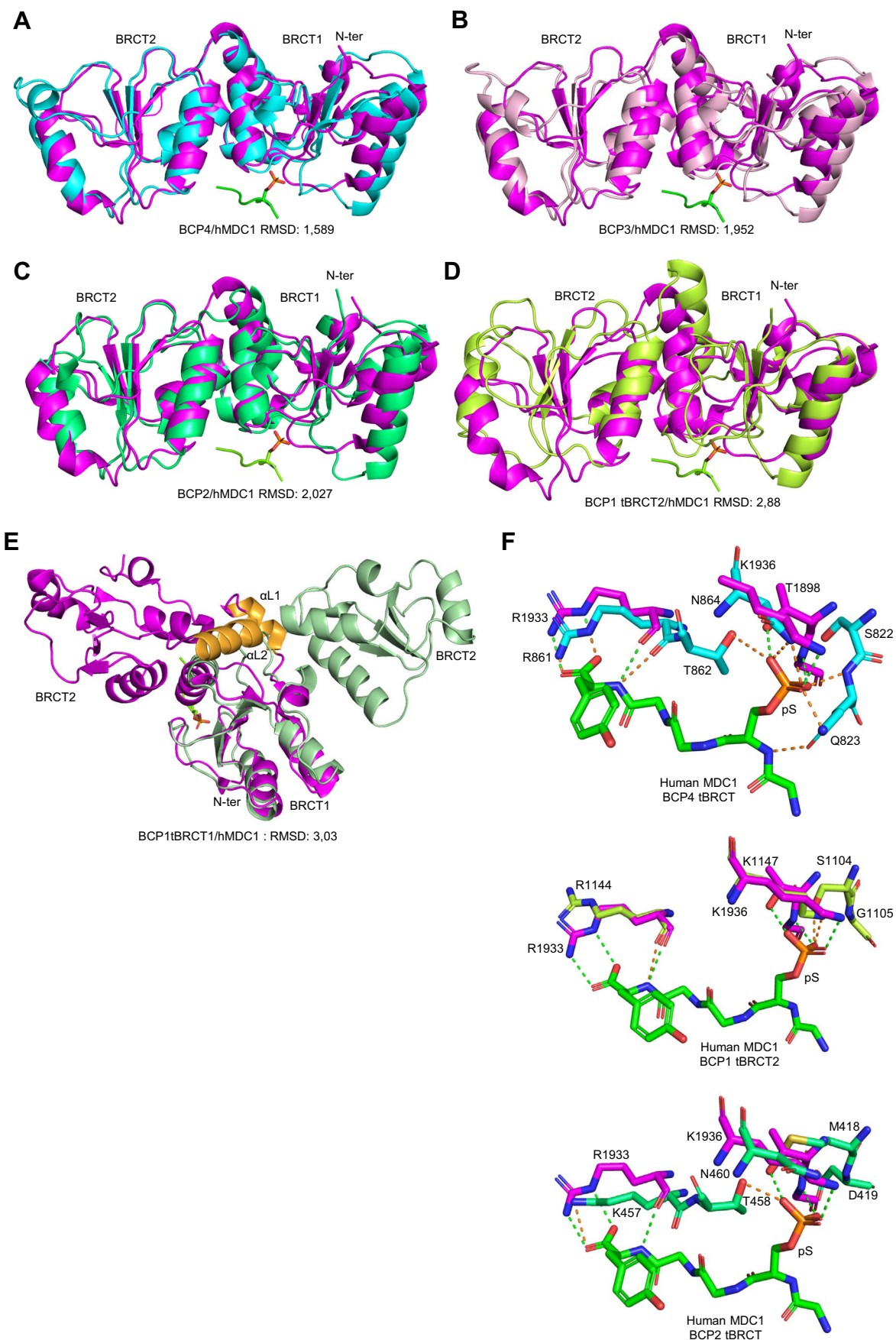

◄  **Figure EV3.  Predicted structures of BCP proteins.**

(**A–E**) AlphaFold2 models of the tBRCT domains of BCP4 (**A**), BCP3 (**B**), BCP2 (**C**), and BCP1 (**D**, **E**) superimposed with a structure of human MDC1 in complex with the phosphorylated C-terminal peptide of H2A.X (Stucki et al, 2005). Note that tBRCT1 of BCP1 overlaps only with MDC1 N-terminal BRCT domain. This is presumably due to the absence of alpha helices (αL1 and αL2, in orange) in tBRCT1 of BCP1 connecting two BRCT domains as indicated in orange. In all panels, human MDC1 tBRCT is purple colored. (**F**) Comparison of interactions of the tBRCT domains of BCP1 and BCP2 with phosphorylated C-terminal peptide of H2A.X. Published (MDC1; Stucki et al, 2005) and AlphaFold2 predicted (BCP1 and BCP2) contacts of amino acids with pSer of H2A.X are indicated respectively with green and orange dotted lines. For a comparison, BCP4 is displayed to indicate reduced abilities of the tBRCT domains of BCP1 and BCP2 to contact pSer of H2A.X. Human MDC1 tBRCT is purple colored.

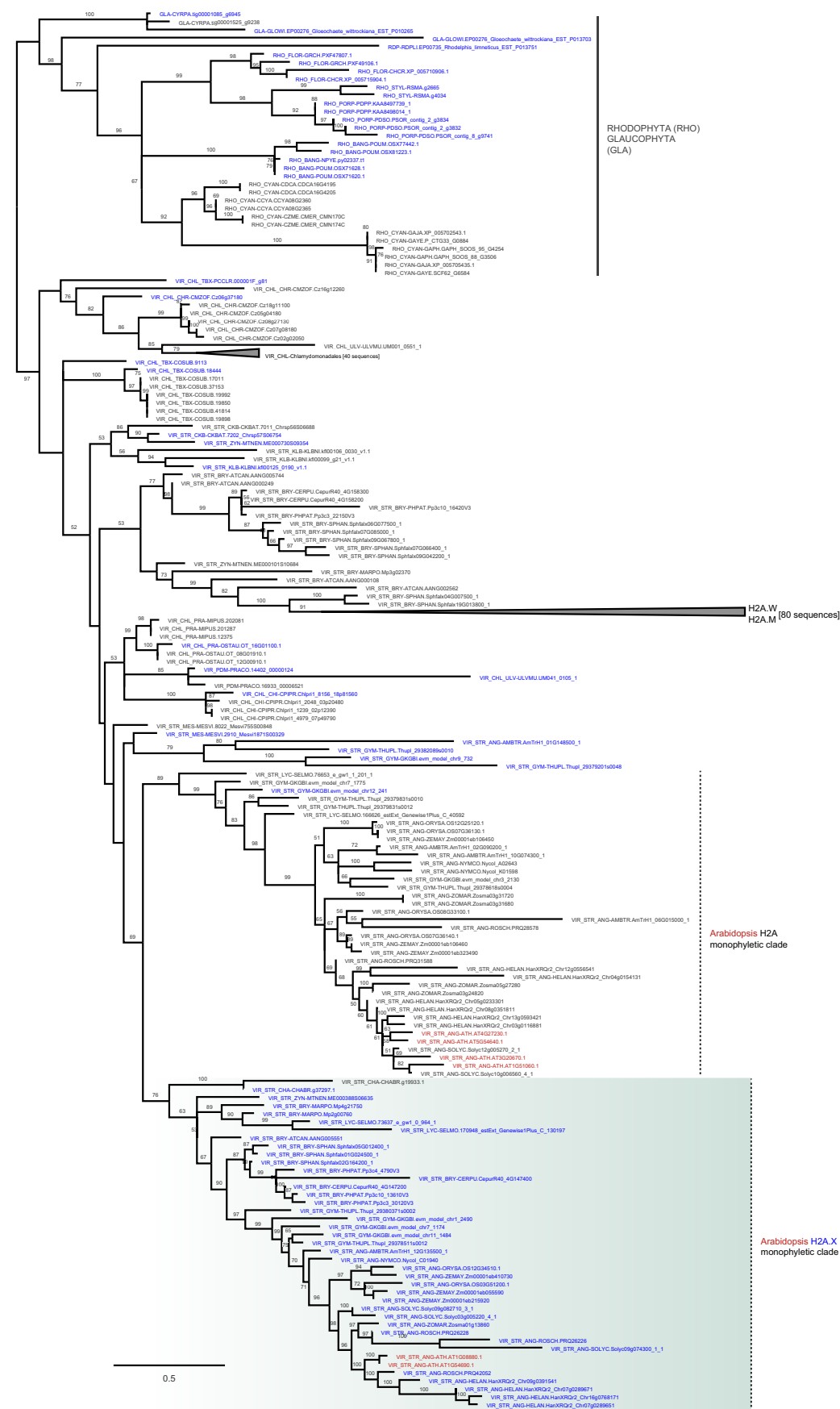

◄ **Figure EV4. Phylogeny of H2A variants from Archaeplastida H2A orthogroup.**

The phylogenetic positions of *Arabidopsis* H2A variants are marked in red. H2A.X sequences with a SQEF/Y motif or SQEF-like (SQ + E/D + F/I/L/V/Y) motif at the C-terminus are highlighted in blue. Note that in green algae, red algae, and glaucophytes H2A and SQEF/Y motif-containing H2As do not form separate clades.

**A**

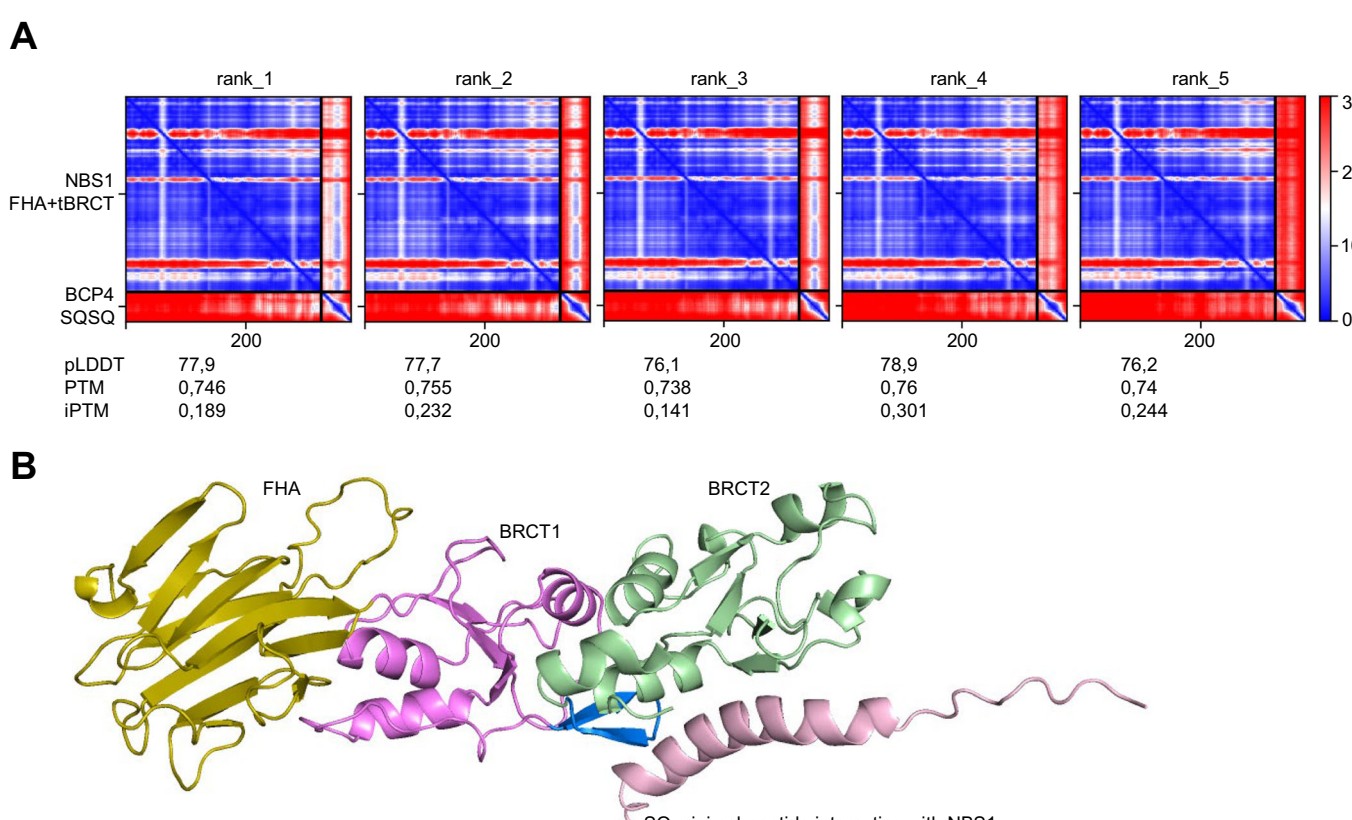

**B**

SQ minimal peptide interacting with NBS1
276-CNVDLGNLSYIG**SQ**EPGEE**SQ**ASALNLVDK**L**ISE**C**RLEFDFEVQADYGRK

**C**

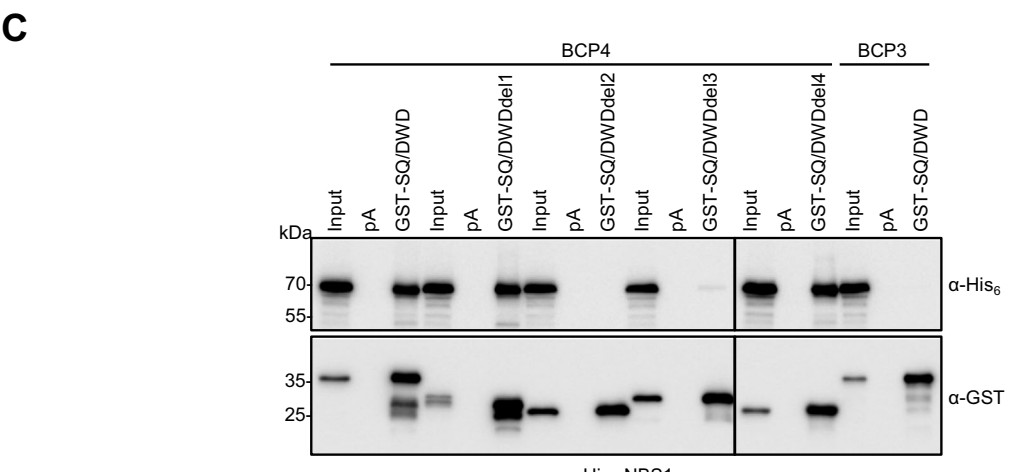

**Figure EV5. Predicted structure of NBS1 and its interaction with SQ/DWD region of BCP4.**

(A) PAE plots of NBS1 (1–324) and BCP4 SQ/DWD (276–325), with calculated predicted local distance difference test (pLDDT), predicted template modeling (PTM), and interface-predicted template modeling (iPTM) scores. (B) AlphaFold2 model of *Arabidopsis* NBS1 in complex with SQ/DWD region of BCP4 (top panel). A sequence of BCP4 minimal peptide interacting with NBS1 is indicated with Leu and Cys residues involved in interaction with NBS1 highlighted in red. Schematic presentation of NBS1 with indicated conserved domains (bottom panel). (C) Co-purification of NBS1 with GST-tagged deletion 4 of BCP4 SQ/DWD motif and its point mutants. Source data are available online for this figure.

