## [Peer Review File · EMBO Reports]

Identification of plants' functional counterpart of the metazoan Mediator of DNA Damage Checkpoint 1

Zdravko Lorkovic, Michael Klingenbrunner, Chung Hyun Cho, and Frédéric Berger

Corresponding author(s): Frédéric Berger (frederic.berger@gmi.oeaw.ac.at)

Review Timeline:

Transfer from Review Commons:	4th Jan 24
Editorial Decision:	5th Feb 24
Revision Received:	13th Feb 24
Accepted:	19th Feb 24

Editor: Esther Schnapp

Transaction Report: This manuscript was transferred to EMBO reports following peer review at Review Commons.

**Review
COMMONS**

Review #1

1. Evidence, reproducibility and clarity:

Evidence, reproducibility and clarity (Required)

MDC1 is a key regulator of DNA damage responses (DDR) in animals. MDC1 has multiple protein domains, in which the BRCT domain binds γ H2A.X. However, plants lack the homolog of MDC1. In this study, the authors found that BCP4 binds γ H2A.X and proposed that BCP4 is a functional counterpart of MDC1, which will greatly enhance our understanding of plant DDR pathway. I have the following concerns.

1. The relationship between BCP3 and BCP4 needs to be clarified. Line 255, the authors mentioned that "we conclude that BCP3 and BCP4 have functional properties as human MDC1". In the Abstract, the authors mentioned that "we identified BCP4 as a candidate ortholog of human MDC1". I am confused about the conclusion. Both BCP3 and BCP4 are or only BCP4 is MDC1? In addition, in BCP3 and BCP4, only their BRCT domains share homology with MDC1. They lack other domains of MDC1. Therefore, "ortholog" may not be an appropriate term. I think "functional counterpart" may be a better term.
2. BCP1-4 all contains tandem BRCT domains. I am wondering whether it is possible to figure out why only BCP3 and BCP4 bind γ H2A.X through sequence analysis. Are there any key residues essential for γ H2A.X binding?
3. Line 183, "On an unrooted phylogenetic tree, these two proteins clustered with MDC1 and PAXIP1 (Figure 1B)". In Figure 1B, MDC1 is closer to BCP3 and BCP4 than PAXIP1 and PAXIP1 is closer to BCP2 than MDC1. If the authors want to include PAXIP1 in Figure 1C, the authors should include BCP2 as well. In the γ H2A.X binding assays, I do not understand why the authors tested BCP1 instead of BCP2.
4. The expression level of BCP1-4 in the mutants need to be examined using qRT-PCR. Especially, for the *bcp3* mutant, which is a weak allele.
5. The authors used "bleomycin" or "zeocin" in different parts. Please be consistent.
6. In Figure 2D, why *bcp2* was not included?
7. Figure 3E and 3F, please indicate the treatments of the upper and lower panels.
8. Line 338, "*bcp1* mutants show reduced homologous recombination rates (Fan et al., 2022; Vladejić et al., 2022; Yu et al., 2023)". The *bcp1* mutant was not reported in Fan et al. paper.
9. Line 40, please add a comma after "In animal". Line 331, please add a comma after "In mammals".

10. Line 123, "only BRCA1 and BARD1 were described in plant lineage". Additional BRCT proteins were described in plants, including XIPI (Nat. Commun. 13:7942), BCP1/DDRM2 (New Phytol. 238:1073-1084; Front. Plant Sci. 13:1023358), and DDRM1 (PNAS, 119: e2202970119).

2. Significance:

Significance (Required)

This study identified BCP4 as a functional counterpart of MDC1, which filled the gap of plant DDR signaling.

3. How much time do you estimate the authors will need to complete the suggested revisions:

Estimated time to Complete Revisions (Required)

(Decision Recommendation)

Between 1 and 3 months

Yes

Review #2

1. Evidence, reproducibility and clarity:

Evidence, reproducibility and clarity (Required)

In this study, Frédéric Berger and colleagues identified BCP4 in *Arabidopsis thaliana* as a potential plant orthologue of vertebrate MDC1. The conclusions are based on both in silico analysis (phylogenetic analysis) and in vitro biochemical and cell biological experiments. BCP4 loss causes sensitivity of DNA damage. Moreover, BCP4 binds to a phosphopeptide derived from the C-terminus of H2AX, via its C-terminal BRCT domains and forms foci in cells exposed to DNA damage, which co-localize with gammaH2AX foci.

****Major comments:****

The conclusions are generally supported by the data, but the evidence presented is still quite limited. For example, it is still possible that BCP4 recruitment to sites of DNA damage is mediated by another protein and not by direct interaction with gammaH2AX. To firmly conclude that BCP4 is an MDC1 orthologue, it is in my opinion essential to perform a (limited) mutagenesis analysis. The key amino acids in the BRCT domains that recognize gammaH2AX need to be mutated and it has to be shown that these mutants are defective for H2AX phosphopeptide binding and are not recruited to sites of DNA damage. Such residues may be tricky to identify, but one obvious candidate would be the Ser residue in beta1 (VLFS motif). In vertebrates, this is a Thr that directly interacts with the phosphate in gammaH2AX. Another possible critical site may be shortly before alpha2 (RTRN motif). In vertebrates, it is RTVK, and the K makes direct contacts with the phosphate in gammaH2AX. This function is perhaps carried out by an R. Structure prediction with alphafold may help to identify the most critical residues

Another critical issue is the introduction of the study. This needs to be revised, because the literature is not correctly cited in several places. For example, the cited paper by Salguero et al., 2019 did not show that the PST repeats of MDC1 constitute a docking site of TP53BP1, but instead, that the PST repeats can bind to chromatin independently of gammaH2AX.

The data are generally well presented and convincing. The only thing that needs to be added is a quantification of the microscopic analysis (e.g. number of foci per cell, or similar).

****Optional:**** it would be interesting to address the question why plants seem to have two MDC1 orthologues. The longer BCP4 and the shorter BCP3. What is the functional difference between those? Do they perhaps distribute functions that are combined in one protein in vertebrate MDC1 on two different proteins?

2. Significance:

Significance (Required)

The strength of the study is the detailed phylogenetic analysis. Also, the biochemistry and cell biology is well done.

Limitations are the lack of evidence that BCP4 carries out functions in the cell (beyond recognising gammaH2AX) that are carried out by MDC1 in vertebrate cells

The study is of great interest to readers working on chromatin responses to DNA damage in plants.

3. How much time do you estimate the authors will need to complete the suggested revisions:

Estimated time to Complete Revisions (Required)

(Decision Recommendation)

More than 6 months

No

Review #3

1. Evidence, reproducibility and clarity:

Evidence, reproducibility and clarity (Required)

****Summary****

The authors set out to find proteins containing BRCT domain to isolate the readers of phosphorylated H2A.X in plants. Using systematic analysis of the BRCT domain proteome, they discovered 21 proteins. Further analysis showed that BCP3 and BCP4 are the ortholog of animal MDC1 and BCP1 is the animal ortholog of PAXIP1. They also extended their work to an evolutionary perspective, finding that BCP1 and BCP4 in plants and PAXIP1 and MDC1 in metazoans evolved independently from a common ancestor. However, this manuscript raises some concerns. Checkout the comments and questions below.

****Major comments:****

1. If you think that BCP3 and BCP4 work as a mediator of DDR, can you show us that those mutants have a defect of DDR? The authors only assessed true leaf developing. Leaf developing is affected by not only DNA damage but also other factor. Therefore, authors should show us additional data showing the BCP mutant lines show defective of DNA damage response.
2. Do you have DNA damage sensitivity data for *bcp3 bcp4* double mutants?
3. Some red algae have H2A.X but don't have BCP4 and BCP1 (Figure 4). In this case, how do they read the phosphorylated H2A.X? Can you discuss the point?
4. L307-L312: I thought that the timing of the appearance of SQEF motif in H2A.X differ from the appearance of BCP4 from Figure 4. Why do you say that the evolution of BCP4 and H2A.X coincides?
5. Some red algae don't have BCP1, BCP4 and H2A.X. How do they transfer the signal to downstream? Do you have any idea about this?
6. Title is not clear to understand. Please change it more suitable one.

****Minor comments:****

7. I think you should show us a schematic representation of BAP1 and PAXIP1 to compare both protein features.
8. L176-L178: Which data support this sentence?
9. L271-L279: There are unreadable characters at "TQx_".

2. Significance:

Significance (Required)

General assessment:

This study give us an idea how organisms have evolved the upstream system of DDR.

Advances:

This study extend the knowledge of DNA damage response in plants.

Audience:

broad and basic research

3. How much time do you estimate the authors will need to complete the suggested revisions:

Estimated time to Complete Revisions (Required)

(Decision Recommendation)

Less than 1 month

Yes

Full Revision

Manuscript number: RC-2023-02012

Corresponding author(s): Frederic, Berger

[Please use this template only if the submitted manuscript should be considered by the affiliate journal as a full revision in response to the points raised by the reviewers.]

*If you wish to submit a preliminary revision with a revision plan, please use our "Revision Plan" template. **It is important to use the appropriate template to clearly inform the editors of your intentions.***

1. General Statements [optional]

This section is optional. Insert here any general statements you wish to make about the goal of the study or about the reviews.

We thank reviewers for useful suggestions and comments on our manuscript which helped to improve and strengthen our conclusions. Our point-by-point answers are below. We have answered most of the points raised by the reviewers and added numerous new experimental data including detailed structural and biochemical analyses that led to support further that BCP4 (and not BCP3) is the plant functional counterpart of MDC1 because in response to DNA damage it binds phosphorylated H2A.X and recruits the MRN complex. In addition, we provide further support to the phylogenetic analysis and evidence for the plant counterpart of PAXIP1. We believe that our revised manuscript which includes a set of new experimental data strongly support our main conclusion that BCP4 is a functional counterpart of metazoan MDC1.

This section is mandatory. Please insert a point-by-point reply describing the revisions that were already carried out and included in the transferred manuscript.

Reviewer #1 (Evidence, reproducibility and clarity (Required)):

MDC1 is a key regulator of DNA damage responses (DDR) in animals. MDC1 has multiple protein domains, in which the BRCT domain binds γ H2A.X. However, plants lack the homolog of MDC1. In this study, the authors found that BCP4 binds γ H2A.X and proposed that BCP4 is a functional counterpart of MDC1, which will greatly enhance our understanding of plant DDR pathway. I have the following concerns.

1. The relationship between BCP3 and BCP4 needs to be clarified. Line 255, the authors mentioned that "we conclude that BCP3 and BCP4 have functional properties as human MDC1".

In the Abstract, the authors mentioned that "we identified BCP4 as a candidate ortholog of human MDC1". I am confused about the conclusion. Both BCP3 and BCP4 are or only BCP4 is MDC1? In addition, in BCP3 and BCP4, only their BRCT domains share homology with MDC1. They lack other domains of MDC1. Therefore, "ortholog" may not be an appropriate term. I think "functional counterpart" may be a better term.

Response: Our analysis emphasizes the fact that human MDC1 is very derived from an ancestral form MDC1 that did not share most domains found in MDC1 from mammals. Because it is still difficult to establish with certainty what the ancestral MDC1 was, we agree that functional counterpart is a more correct term, so we changed this accordingly throughout the manuscript.

2. BCP1-4 all contains tandem BRCT domains. I am wondering whether it is possible to figure out why only BCP3 and BCP4 bind γ H2A.X through sequence analysis. Are there any key residues essential for γ H2A.X binding?

Response: We used AlphaFold models of tBRCT domain of BCP1, BCP2, BCP3, and BCP4. While in AlphaFold models the tBRCT domain of each BCP protein largely overlaps with a structure of human MDC1 tBRCT domain, only the tBRCT domain of BCP3 and BCP4 are predicted to make contacts with γ H2A.X similar to that of human MDC1. Although residues that are involved are not fully conserved between BCP3/4 and human MDC1 we obtain in vitro data supporting that the interaction of BCP4 is mediated by a comparable pocket of three key residues that contact the phosphate group of γ H2A.X. See also answers to comments of Referee #2 and new Figures 3 and 4, corresponding description on page 8-9, and Supporting figure 3.

3. Line 183, "On an unrooted phylogenetic tree, these two proteins clustered with MDC1 and PAXIP1 (Figure 1B)". In Figure 1B, MDC1 is closer to BCP3 and BCP4 than PAXIP1 and PAXIP1 is closer to BCP2 than MDC1. If the authors want to include PAXIP1 in Figure 1C, the authors should include BCP2 as well. In the γ H2A.X binding assays, I do not understand why the authors tested BCP1 instead of BCP2. In Figure 2D, why bcp2 was not included?

Response: We created a new alignment for Figure 1C including BCP2 tBRCT domain and the tree that includes all BCP BRCT domain (Figure 1D) does support a close relation between MDC1 and BCP3 and 4 and PAXIP1 and BCP1. As we stated on page 5-6 lines 175-178, BCP2, also contains acetyltransferase domain, which is unique for plant BCP2 protein. Based on its domain organization, BCP2 was not considered as a candidate for MDC1 homolog, and we did not perform mutant complementation. This is why after our initial analysis of *bcp* mutants (DNA damage sensitivity, formation of γ H2A.X, and phylogeny), and based on similarities with MDC1 and PAXIP1 we focused on *bcp1*, 3, and 4 mutants and the corresponding proteins. The function of BCP2 remains to be investigated, but this is out of the scope of this manuscript that is primarily dedicated to find the functional counterparts of MDC1 and PAXIP1.

4. The expression level of BCP1-4 in the mutants need to be examined using qRT-PCR. Especially, for the bcp3 mutant, which is a weak allele.

Response: We did not perform this experiment, because it was done in Vladejic et al., 2022 and expression data are available from various genomic dataset on TAIR.

5. The authors used "bleomycin" or "zeocin" in different parts. Please be consistent.

Response: We consistently use Bleomycin for treatment of seedlings followed by western blotting and Zeocin for true leaf assay. These two agents produce DNA double strand brakes in similar ways, and we could show previously that levels of γ H2A.X and γ H2A.W.7 are similar when using these two agents (Rosa M, Mittelsten Scheid O *Bio. Protoc.* 4:e1093. doi: 10.21769/BioProtoc.1093; Lorkovic et al., *Curr Biol.* 2017, doi: 10.1016/j.cub.2017.03.002). Zeocin was chosen for true leaf assays because we observe lower variation between batches and biological repeats compared with bleomycin.

7. Figure 3E and 3F, please indicate the treatments of the upper and lower panels.

Response: Thank you for pointing this out. This has been indicated in the corresponding legend of the new Figure 3 A - C.

8. Line 338, "bcp1 mutants show reduced homologous recombination rates (Fan et al., 2022; Vladejić et al., 2022; Yu et al., 2023)". The bcp1 mutant was not reported in Fan et al. paper.

Response: This sentence has been changed to accurately describe data in each of the mentioned papers.

9. Line 40, please add a comma after "In ". Line 331, please add a comma after "In mammals". animal

Response: This has been corrected.

10. Line 123, "only BRCA1 and BARD1 were described in plant lineage". Additional BRCT proteins were described in plants, including XIP1 (*Nat. Commun.* 13:7942), BCP1/DDRM2 (*New Phytol.* 238:1073-1084; *Front. Plant Sci.* 13:1023358), and DDRM1 (*PNAS*, 119: e2202970119).

Response: This sentence refers to known BRCT domain mediator/effector proteins. From the published data about XIP1, BCP1/DDRM2, and DDRM1, it is not possible to assign these functions to proteins in question. Nevertheless, we changed this sentence to avoid ambiguous interpretation and we later in the text introduce XIP1, BCP1/DDRM2, and DDRM1 proteins as needed.

Reviewer #1 (Significance (Required)):

This study identified BCP4 as a functional counterpart of MDC1, which filled the gap of plant DDR signaling.

Reviewer #2 (Evidence, reproducibility and clarity (Required)):

In this study, Frédéric Berger and colleagues identified BCP4 in *Arabidopsis thaliana* as a potential plant orthologue of vertebrate MDC1. The conclusions are based on both in silico analysis (phylogenetic analysis) and in vitro biochemical and cell biological experiments. BCP4 loss causes sensitivity of DNA damage. Moreover, BCP4 binds to a phosphopeptide derived from the C-terminus of H2AX, via its C-terminal BRCT domains and forms foci in cells exposed to DNA damage, which co-localize with gammaH2AX foci.

Major comments:

The conclusions are generally supported by the data, but the evidence presented is still quite limited. For example, it is still possible that BCP4 recruitment to sites of DNA damage is mediated by another protein and not by direct interaction with gammaH2AX. To firmly conclude that BCP4 is an MDC1 orthologue, it is in my opinion essential to perform a (limited) mutagenesis analysis. The key amino acids in the BRCT domains that recognize gammaH2AX need to be mutated and it has to be shown that these mutants are defective for H2AX phosphopeptide binding and are not recruited to sites of DNA damage. Such residues may be tricky to identify, but one obvious candidate would be the Ser residue in beta1 (VLFS motif). In vertebrates, this is a Thr that directly interacts with the phosphate in gammaH2AX. Another possible critical site may be shortly before alpha2 (RTRN motif). In vertebrates, it is RTVK, and the K makes direct contacts with the phosphate in gammaH2AX. This function is perhaps carried out by an R. Structure prediction with alphaFold may help to identify the most critical residues

Response: We thank the reviewer for these suggestions. We used AphaFold to predict structures of tBRCT domains of all BCPproteins and compared them with structure of human MDC1 in complex with gamaH2A.X peptide. Based on these analyses we performed mutagenesis of critical amino acids in BCP4 based on their predicted interaction and their conservation. We showed that mutations of critical residues reduced or almost completely abolished binding of BCP4 to γH2A.X. These data are now part of the new Figure 4. See also corresponding description on page 8-9.

In addition we provide genetic data that show that the foci formation of BCP4 depends on H2A.X (new Fig 3B and C). We did not attempt genetic complementation experiments with these mutants because it would take nine months to obtain stable transgenic plant lines expressing various mutant versions of BCP4 and the limitation of *Arabidopsis* transgenesis does not allow to control precisely the expression of transgenes, which could cause a difficult interpretation in this particular case.

Another critical issue is the introduction of the study. This needs to be revised, because the literature is not correctly cited in several places. For example, the cited paper by Salguero et al., 2019 did not show that the PST repeats of MDC1 constitute a docking site of TP53BP1, but instead, that the PST repeats can bind to chromatin independently of gammaH2AX.

Response: We thank the reviewer for spotting this mistake. We carefully checked all references and corrected all wrongly associated ones or used original reports instead of reviews. Also, we did re-write some parts of the Introduction as referee #1 also asked for some clarification.

The data are generally well presented and convincing. The only thing that needs to be added is a quantification of the microscopic analysis (e.g. number of foci per cell, or similar).

Response: We quantified the foci number in all mutants reported in Figure 2C. These data are now included in the new Figure 2D.

Optional: it would be interesting to address the question why plants seem to have two MDC1 orthologues. The longer BCP4 and the shorter BCP3. What is the functional difference between those? Do they perhaps distribute functions that are combined in one protein in vertebrate MDC1 on two different proteins?

Response: Thank you for prompting us to address this outstanding question. We now provide evidence supporting that only BCP4 is a functional counterpart of MDC1. We show that a specific region of BCP4 but not BCP3 is able to interact with NBS1 of the MRN complex (see new Figure 6). Also, BCP3 is missing the N-terminal TQxφ repeats present in BCP4. Although the function of these repeats is unknown at this point, these data together suggest some functional diversification between BCP3 and BCP4. We mention this on page 11, lines 372-374.

Reviewer #2 (Significance (Required)):

The strength of the study is the detailed phylogenetic analysis. Also, the biochemistry and cell biology is well done.

Limitations are the lack of evidence that BCP4 carries out functions in the cell (beyond recognising gammaH2AX) that are carried out by MDC1 in vertebrate cells

Response: We thank the reviewer for pointing out this important point. To address it we performed pull-down assays with TQxφ and SQ/DWD regions of BCP4 with NBS1 and found that Arabidopsis NBS1 interacts with the SQ/DWD region, and that this interaction is mediated by FHA+tBRCT of NBS1. Based on AlphaFold prediction, we performed further deletion and point mutation analysis of the SQ/DWD region and defined that the binding of NBS1 requires an alpha-helix comprising sequence that is not conserved in BCP3. So, we concluded that a sequence specific of BCP4 (not in BCP3) is capable of recruiting the MRN subunit NBS1. At this point we could not demonstrate this in vivo by analyzing NBS1 foci in BCP4 mutant background. Unfortunately, commercial antibodies for plant NBS1 or other subunits of the MRN complex are not available, and to get transgenic plants expressing fluorescent protein tagged NBS1 would require a period much longer than the time for reasonable revisions of a manuscript. Nevertheless, our in vitro interaction data strongly argue for BCP4 having function in binding MRN complex as human MDC1, although the mode of interaction of BCP4 with NBS1 is different from that of human MDC1 and NBS1.

Please see the new Figure 6 and corresponding description on page 11-12.

The study is of great interest to readers working on chromatin responses to DNA damage in plants.

Reviewer #3 (Evidence, reproducibility and clarity (Required)):

Summary

The authors set out to find proteins containing BRCT domain to isolate the readers of phosphorylated H2A.X in plants. Using systematic analysis of the BRCT domain proteome, they discovered 21 proteins. Further analysis showed that BCP3 and BCP4 are the ortholog of animal MDC1 and BCP1 is the animal ortholog of PAXIP1. They also extended their work to an evolutionary perspective, finding that BCP1 and BCP4 in plants and PAXIP1 and MDC1 in metazoans evolved independently from a common ancestor. However, this manuscript raises some concerns. Checkout the comments and questions below.

Major comments:

1) If you think that BCP3 and BCP4 work as a mediator of DDR, can you show us that those mutants have a defect of DDR? The authors only assessed true leaf developing. Leaf developing is affected by not only DNA damage but also other factor. Therefore, authors should show us additional data showing the BCP mutant lines show defective of DNA damage response.

Response: The “true leaf assay” is a classical assay for testing plant mutants for DNA damage sensitivity (Rosa M, Mittelsten Scheid O *Bio. Protoc.* 4:e1093. doi: 10.21769/BioProtoc.1093). If DNA damage occurs and is not efficiently repaired, meristematic cells in shoot meristem are arrested and do not divide, hence plants do not produce the first pair of “true” leaves after cotyledons expand. In this assay cotyledons open and grow normally as they are already fully determined and do not undergo any cell division after seed germination.

In this assay the treated WT seedlings also show a reduction of the number of plants with true leaves as compared with untreated WT (100%). Furthermore, WT and mutant seedlings develop normally and comparably without Zeocin induced DNA damage.

2) Do you have DNA damage sensitivity data for *bcp3 bcp4* double mutants?

Response: We obtained *bcp3bcp4* double mutant and tested it for DNA damage sensitivity. The double mutant is slightly more sensitive than *bcp4* single mutant, but not as sensitive as H2A.X mutant. The reason for this is presumably the nature of the *bcp3* mutant allele available, with a T-DNA insertion located in the 5'-UTR with some residual expression of BCP3 protein as reported by Vladejic et al., 2022. We did not feel that this would improve the manuscript, so we did not include this data. To obtain a new mutant allele would take time and work beyond the reasonable time required for revision. In addition, since we show that the functional counterpart of MDC1 is BCP4, we did not think that it is relevant to pursue further the characterization of the function of BCP3 in the context of this manuscript.

3) Some red algae have H2A.X but don't have BCP4 and BCP1 (Figure 4). In this case, how do they read the phosphorylated H2A.X? Can you discuss the point?

Response: Actually, most red algae do not even have H2A.X. At this point we do not have data that could answer this question and it is difficult to make any prediction about this. Analysis of DDR system in red algae is totally beyond the scope of the current manuscript. See also answer to comment #5.

4) L307-L312: I thought that the timing of the appearance of SQEF motif in H2A.X differ from the appearance of BCP4 from Figure 4. Why do you say that the evolution of BCP4 and H2A.X coincides?

Response: we thank the Reviewer for pointing out the need for clarification.

Histone H2A with a C-terminal SQEF/Y motif is categorized as H2A.X that distinguishes this variant from H2A.Z (not discussed here) and H2A itself. In Archaeplastida many algal species possess either H2A or H2A.X. Only in streptophytes the ancestral gene duplicated leading to neofunctionalization of both H2A and H2A.X and in this case H2A.X form a **monophyletic** clade. The evolution of BPC1 and 4 are slightly posterior or coincident with this neofunctionalized H2A.X variant, suggesting co-evolution in streptophytes.

5) Some red algae don't have BCP1, BCP4 and H2A.X. How do they transfer the signal to downstream? Do you have any idea about this?

Response: To address this interesting question we re-analyzed BRCT domain proteome of the red algae and again could not find any protein containing features of BCP4 present in green algae and land plants or in Opisthokont MDC1.

We did find that red algae without MDC1 do encode MRE11, RAD50 but not NBS1. Also, components of non-homologous end joining DNA repair pathway, Ku70 and Ku80 are conserved in these organisms. So, how some red algae cope with DNA damage remains enigmatic. Similarly unicellular red algae do not have the classical autophagy pathway. This is the result of the very strong genome reduction (< 20Mb). Red algae likely involve alternative pathways, yet to be identified, in recruiting DNA repair machinery. As answering this question was out of the scope of this work, we did not include any further comments on this issue.

6) Title is not clear to understand. Please change it more suitable one.

Response: Thanks for this comment. We did change title of the manuscript to avoid ambiguity.

Minor comments:

6) I think you should show us a schematic representation of BAP1 and PAXIP1 to compare both protein features.

Response: We added schematic presentation of PAXIP1 to Supporting Figure 2B.

7) L176-L178: Which data support this sentence?

Response: The sentence in question: "BCP1 has two tBRCT domains positioned at the N- and C-terminus and a so far unrecognized C-terminal PHD finger which is present in all plant lineages except Brassicaceae (Supporting Figure S1A and S2A)."

Response: Our data presented on Supporting Figure S1A (schematic presentation of BCP1 protein with indicated PHD finger consensus sequence) and S2A and Source data (alignment of

Full Revision

PHD fingers in BCP1 in flowering plants, non-flowering land plants and multicellular green algae) clearly demonstrate the presence of a C-terminal PHD finger in BCP1 except in Brassicaceae. These can also be seen in the full complement of BCP1 sequences that are available in Source data.

8) L271-L279: There are unreadable characters at "TQx_".

Response: This very likely appeared during conversion into PDF file. We fixed this now.

Reviewer #3 (Significance (Required)):

Significance:

General assessment:

This study give us an idea how organisms have evolved the upstream system of DDR.

Advances:

This study extend the knowledge of DNA damage response in plants.

Audience:

broad and basic research

Dear Dr. Berger,

Thank you for the submission of your revised manuscript. We have now received the enclosed reports from the referees. Referee 3 still has a few more suggestions that I would like you to address and incorporate before we can proceed with the official acceptance of your manuscript.

A few editorial requests will also need to be addressed:

- Please reduce the number of keywords to 5.
- Please remove the author credits from the ms file. All credits need to be entered online during ms submission.
- Please correct the reference format to the EMBO reports style - DOIs should only be used for preprints and datasets that have not been published yet; and please double-check whether "et al" is used after 10 author names. The EMBO reports reference style is also in EndNote.
- Please co-submit a completed author checklist, which you can download from our author guidelines . The completed author checklist will also be part of the transparent peer-review process file.
- The funding info in the ms does not match what is entered during ms submission online, please correct. All funding info must be listed also during online ms submission.
- Please upload all main and EV (expanded view) figures as one individual file per figure. The EV figure legends need to be listed after the main figure legends in the main ms file. You can find more information about our figure types in our guide to authors online.
- Please move Table 1 to the methods section.
- You uploaded 3 pdf files with source data as Suppl. Material. Please upload source data as source data files. Hannah will also contact you with more information regarding the source data.
- Please address the following comments by our data editors:
 1. Please note that a separate 'Data Information' section is required in the legends of figures 2a, d; 3a-f; 5a-c.
 2. Please note that the box plot needs to be defined in terms of minima, maxima, centre, bounds of box and whiskers, and percentile in the legend of figure 2d.
 3. Please note that information related to n is missing in the legends of figures 2a-e.

I would like to suggest a few minor changes to the title and abstract. Please let me know whether you agree with the following:

Identification of plants' functional counterpart of the metazoan Mediator of DNA Damage Checkpoint 1

Induction of DNA damage triggers rapid phosphorylation of histone H2A.X (γ H2A.X). In animals, mediator of DNA damage checkpoint 1 (MDC1) binds γ H2A.X through a tandem BRCA1 carboxyl-terminal (tBRCT) domain and mediates recruitment of downstream effectors of the DNA damage response (DDR). However, readers of this modification in plants have remained elusive. We show that from the Arabidopsis BRCT domain proteome, BCP1-4 proteins with tBRCT domains are involved in DDR. Through its tBRCT domain BCP4 binds γ H2A.X in vitro and localizes to DNA damage-induced foci in an H2A.X-dependent manner. BCP4 also contains a domain that interacts directly with NBS1 and thus acts as a functional counterpart of MDC1. We also show that BCP1, that contains two tBRCT domains, co-localizes with γ H2A.X but it does not bind γ H2A.X, suggesting functional similarity with human PAXIP1. A phylogenetic analysis supports that PAXIP1 and MDC1 in metazoa and their plant counterparts evolved independently from common ancestors with tBRCT domains. Collectively, our study reveals missing components and provides mechanistic and evolutionary insights into plant DDR.

EMBO press papers are accompanied online by A) a short (1-2 sentences) summary of the findings and their significance, B) 2-3 bullet points highlighting key results and C) a synopsis image that is exactly 550 pixels wide and 200-600 pixels high (the height is variable). You can either show a model or key data in the synopsis image. Please note that text needs to be readable at the final size. Please send us this information along with the final manuscript.

Referee #1:

The authors have performed additional experiments to support the conclusion. I have no other concerns.

Referee #3:

This review addresses the revised version of the manuscript submitted to Review Commons by the same authors last year. My previous concerns have been mostly addressed. Specifically, much more evidence has been collected to support the conclusion that BCP4 is a plant MDC1 orthologue. I particularly liked the AlphaFold2 guided site directed mutagenesis approach. The manuscript would have benefitted from additional quantitative assessment of the interaction between BCP4 tBRCT domains and γ H2AX phosphopeptides, e.g. by fluorescence polarisation. However, this is not essential in my opinion.

Figure 6 is totally new and identifies a putative NBS1 interaction site in BCP4. This part of the story is still underdeveloped. While AF2-guided structural modelling and biochemical interaction studies suggest that plants have evolved a different MDC1 interaction mechanism with NBS1, I found it a bit peculiar that the plant NBS1 N-terminal regions still contained the FHA/BRCT domains present in vertebrate NBS1, but would only use BRCT2 to interact with MDC1(BCP4). This is quite unlikely given that vertebrate MDC1 interacts with both FHA domain and tBRCT domains. It is therefore likely that there are other regions in BCP4 that interact with NBS1.

It is also a pity that the authors did not include any cell biology data in this part of the story. For example, it would have been nice to see if BCP4 and NBS1 co-localize in foci and if deletion of BCP4 leads to a defect in MRN foci as does loss of MDC1 in vertebrate cells.

As a minimal revision necessary for acceptance, I would suggest to include quality criteria for the AF2 multimer models. Currently, two parameters are mostly used in the field: pdockq and average interface pae (see e.g. predictomes.org).

Referee #1:

The authors have performed additional experiments to support the conclusion. I have no other concerns.

Referee #3:

This review addresses the revised version of the manuscript submitted to Review Commons by the same authors last year. My previous concerns have been mostly addressed. Specifically, much more evidence has been collected to support the conclusion that BCP4 is a plant MDC1 orthologue. I particularly liked the AlphaFold2 guided site directed mutagenesis approach. The manuscript would have benefitted from additional quantitative assessment of the interaction between BCP4 tBRCT domains and γ H2AX phosphopeptides, e.g. by fluorescence polarisation. However, this is not essential in my opinion.

Figure 6 is totally new and identifies a putative NBS1 interaction site in BCP4. This part of the story is still underdeveloped. While AF2-guided structural modelling and biochemical interaction studies suggest that plants have evolved a different MDC1 interaction mechanism with NBS1, I found it a bit peculiar that the plant NBS1 N-terminal regions still contained the FHA/BRCT domains present in vertebrate NBS1, but would only use BRCT2 to interact with MDC1(BCP4). This is quite unlikely given that vertebrate MDC1 interacts with both FHA domain and tBRCT domains. It is therefore likely that there are other regions in BCP4 that interact with NBS1.

It is also a pity that the authors did not include any cell biology data in this part of the story. For example, it would have been nice to see if BCP4 and NBS1 co-localize in foci and if deletion of BCP4 leads to a defect in MRN foci as does loss of MDC1 in vertebrate cells.

As a minimal revision necessary for acceptance, I would suggest to include quality criteria for the AF2 multimer models. Currently, two parameters are mostly used in the field: pdockq and average interface pae (see e.g. predictomes.org).

Rev_Com_number: RC-2023-02012

New_manu_number: EMBOR-2024-58742V1

Corr_author: Berger

Title: Identification of plants functional counterparts of the metazoan Mediator of DNA Damage Checkpoint 1

Rev_Com_number: RC-2023-02012
New_manu_number: EMBOR-2024-58742V1

Dear Esther,

Thank you for your feedback on our revised manuscript and supportive comments.

We are now addressing a final revised version addressing the following editorial requests

- We reduced the number of keywords to 5.
- We removed the author credits from the ms file and placed them online.
- The reference format was adjusted to the EMBO reports style
- We are co-submitting a completed author checklist.
- The funding listed in the manuscript now match the online data.
- Please upload all main and EV (expanded view) figures as one individual file per figure. The EV figure legends need to be listed after the main figure legends in the main ms file. You can find more information about our figure types in our guide to authors online.
- Table 1 was moved to the methods section.
- Source data was correctly placed to source data files.
- A separate 'Data Information' section is placed in the legends of figures 2a, d; 3a-f; 5a-c.
- The box plot is now defined in the legend of figure 2d.
- n was defined in the legends of figures 2a-e.
- We thank you for the suggested changes to the title and abstract and agree
- We included A) a short (1-2 sentences) summary of the findings and their significance, B) 2-3 bullet points highlighting key results and C) a synopsis image that is exactly 550 pixels wide and 200-600 pixels high (the height is variable). You can either show a model or key data in the synopsis image.
- You will find specific replies to reviewers comments below.

Best regards,
Fred

Referee #3:

This review addresses the revised version of the manuscript submitted to Review commons by the same authors last year.

My previous concerns have been mostly addressed. Specifically, much more evidence has

been collected to support the conclusion that BCP4 is a plant MDC1 orthologue. I particularly liked the AlphaFold2 guided site directed mutagenesis approach. The manuscript would have benefitted from additional quantitative assessment of the interaction between BCP4 tBRCT domains and γ H2AX phosphopeptides, e.g. by fluorescence polarisation. However, this is not essential in my opinion.

Figure 6 is totally new and identifies a putative NBS1 interaction site in BCP4. This part of the story is still underdeveloped. While AF2-guided structural modelling and biochemical interaction studies suggest that plants have evolved a different MDC1 interaction mechanism with NBS1, I found it a bit peculiar that the plant NBS1 N-terminal regions still contained the FHA/BRCT domains present in vertebrate NBS1, but would only use BRCT2 to interact with MDC1(BCP4). This is quite unlikely given that vertebrate MDC1 interacts with both FHA domain and tBRCT domains. It is therefore likely that there are other regions in BCP4 that interact with NBS1.

At this point our data show that interaction of NBS1 and BCP4 is mediated by the second BRCT domain. We agree that this is somehow unexpected considering conservation of the FHA-tBRCT module between plants and animals. However, plant MDC1 (BCP4) does not contain SDTD motifs that are recognized by human NBS1. So, the existence of different modes of interactions may not be surprising.

Nevertheless, we included following statement on page 13, lines 412-417: Although our data strongly support interaction of NBS1 and BCP4, we cannot exclude the possibilities (i) that the SQ dipeptides upstream of the NBS1 binding site are phosphorylated upon induction of the DNA damage and thereby modulate NBS1 binding as well as (ii) the existence of other interaction sites which would involve FHA domain of BCP4 as described for human NBS1 (Lloyd et al, 2009; Williams et al, 2009; Hari et al, 2010).

It is also a pity that the authors did not include any cell biology data in this part of the story. For example, it would have been nice to see if BCP4 and NBS1 co-localize in foci and if deletion of BCP4 leads to a defect in MRN foci as does loss of MDC1 in vertebrate cells.

At this point we could not demonstrate this in vivo by analyzing NBS1 foci in BCP4 mutant background. Unfortunately, commercial antibodies for plant NBS1 or other subunits of the MRN complex are not available, and to get transgenic plants expressing fluorescent protein tagged NBS1 would require a period much longer than the time for reasonable revisions of a manuscript. Nevertheless, our in vitro interaction data strongly argue for BCP4 having function in binding MRN complex as human MDC1, although the mode of interaction of BCP4 with NBS1 seems to be different from that of human MDC1 and NBS1.

Also, we address this at the end of Discussion section with following sentence page 13, lines 437-438: It will also be important to monitor dynamics of BCP4, BCP1, NBS1 (MRN) foci formation in genetic backgrounds depleted of components of plant DDR.

As a minimal revision necessary for acceptance, I would suggest to include quality criteria for the AF2 multimer models. Currently, two parameters are mostly used in the field: pdockq and average interface pae (see e.g. predictomes.org).

We included PAE plots and corresponding iPTM values into Figure EV5A.

Dr. Frédéric Berger
Gregor Mendel Institute (GMI)
Austrian Academy of Sciences, Vienna Biocenter
Dr. Bohr Gasse 3, 1030
Vienna 1030
Austria

Dear Dr. Berger,

I am very pleased to accept your manuscript for publication in the next available issue of EMBO reports. Thank you for your contribution to our journal.

Yours sincerely,

Rev_Com_number: RC-2023-02012

New_manu_number: EMBOR-2024-58742V2

Corr_author: Berger

Title: Identification of plants' functional counterpart of the metazoan Mediator of DNA Damage Checkpoint 1